# Global reorganisation of *cis*-regulatory units upon lineage commitment of human embryonic stem cells

Paula Freire-Pritchett[1†], Stefan Schoenfelder[1†], Csilla Várnai[1], Steven W Wingett[1], Jonathan Cairns[1], Amanda J Collier[2,3], Raquel García-Vílchez[2], Mayra Furlan-Magaril[1‡], Cameron S Osborne[4], Peter Fraser[1], Peter J Rugg-Gunn[2,3*], Mikhail Spivakov[1*]

[1]Nuclear Dynamics Programme, Babraham Institute, Cambridge, United Kingdom; [2]Epigenetics Programme, Babraham Institute, Cambridge, United Kingdom; [3]Wellcome Trust – Medical Research Council Cambridge Stem Cell Institute, University of Cambridge, Cambridge, United Kingdom; [4]Department of Genetics and Molecular Medicine, King's College London School of Medicine, London, United Kingdom

*For correspondence: peter.rugg-gunn@babraham.ac.uk (PJR-G); mikhail.spivakov@babraham.ac.uk (MS)

[†]These authors contributed equally to this work

Present address: [‡]Instituto de Fisiología Celular, Universidad Nacional Autónoma de México, Mexico City, Mexico

Competing interests: The authors declare that no competing interests exist.

**Abstract** Long-range *cis*-regulatory elements such as enhancers coordinate cell-specific transcriptional programmes by engaging in DNA looping interactions with target promoters. Deciphering the interplay between the promoter connectivity and activity of *cis*-regulatory elements during lineage commitment is crucial for understanding developmental transcriptional control. Here, we use Promoter Capture Hi-C to generate a high-resolution atlas of chromosomal interactions involving ~22,000 gene promoters in human pluripotent and lineage-committed cells, identifying putative target genes for known and predicted enhancer elements. We reveal extensive dynamics of *cis*-regulatory contacts upon lineage commitment, including the acquisition and loss of promoter interactions. This spatial rewiring occurs preferentially with predicted changes in the activity of *cis*-regulatory elements and is associated with changes in target gene expression. Our results provide a global and integrated view of promoter interactome dynamics during lineage commitment of human pluripotent cells.

## Introduction

Cell fate decisions are associated with profound changes in chromatin organisation, which underlie the activation of lineage-specific and the silencing of lineage-inappropriate genes (*Buecker and Wysocka, 2012*; *Bulger and Groudine, 2010*; *Calo and Wysocka, 2013*; *Hallikas et al., 2006*; *Ong and Corces, 2012*). *Cis*-regulatory elements such as transcriptional enhancers play a key role in this process by integrating regulatory inputs from intrinsic and extracellular cues, and by mediating the recruitment of core activator and repressor complexes (*Pennacchio et al., 2013*; *Shlyueva et al., 2014*; *Spitz and Furlong, 2012*). The definition of chromatin signatures has enabled the genome-wide identification of enhancer elements across multiple human cell types (*ENCODE Project Consortium, 2012*; *Heintzman et al., 2007, 2009*; *Pennacchio et al., 2006*; *Rada-Iglesias et al., 2011*; *Kundaje et al., 2015*). Chromatin states can also provide a robust read-out of *cis*-regulatory activity associated with poised and active enhancers and have been used to show that widespread changes in enhancer position and activity occur upon cell fate decisions such as the lineage commitment of pluripotent cells (*Creyghton et al., 2010*; *Rada-Iglesias et al., 2011*; *Zentner et al., 2011*).

**eLife digest** Virtually every cell in the body contains the same set of DNA, which encodes thousands of genes. The activities of these genes vary between different types of cells and at different points in time. As a result, our DNA contains a complex array of molecular switches that instruct genes to switch on and off at the right time and in the right cells. These molecular switches, termed regulatory elements, are often a long way away from the genes that they control, and this can make it difficult to find out which switch controls which genes.

DNA is made up of several different building blocks known as bases and the order of these bases encodes specific information about the gene. Every human cell contains approximately two meters of DNA, which is highly folded in the cell nucleus. This three-dimensional folding allows regions that are far apart on the DNA thread to physically contact each other. To reach the genes they control, regulatory elements form loops on the DNA that are near-impossible to predict from looking at the sequence of bases alone. Mapping the locations of these loops can reveal the hidden circuitry within our DNA and help us to understand how unwanted changes (mutations) within regulatory elements may cause disease.

Freire-Pritchett, Schoenfelder et al. set out to reveal how loops between genes and their regulatory elements change as the stem cells specialise into immature brain cells. The experiments show that the pattern of DNA loops is extensively altered after the stem cells specialise into brain cells, that is, some loops are lost and new ones form. Furthermore, the regulatory elements themselves were often toggled between "on" and "off" states. These two processes tend to occur together, so that new loops are formed at the same time as the switch is activated.

Freire-Pritchett, Schoenfelder et al. also show that individual genes are often connected to many different regulatory elements. The next step is to understand how these multiple connections work together to coordinate gene activity, and whether this information could be used to control how stem cells specialise. This knowledge may lead to the development of stem cell-based therapies for stroke, Parkinson's disease and other conditions.

*Cis*-regulatory elements are often considerable distances away from their target gene promoters and may not control their nearest genes (*Carvajal et al., 2001*; *Jeong et al., 2006*; *Marinić et al., 2013*; *Pennacchio et al., 2006*; *Ruf et al., 2011*; *Sagai et al., 2005*; *Spitz et al., 2003*). It is generally accepted that this long-range action is facilitated by DNA-looping interactions (*Pennacchio et al., 2013*; *Shlyueva et al., 2014*). However, specific determinants of chromosomal interactions are still not fully understood, which presents challenges for the high-confidence prediction of regulatory interactions from sequence and epigenetic information (*Mora et al., 2016*; *Roy et al., 2015*; *Shlyueva et al., 2014*; *Whalen et al., 2016*). As a result, the target genes of most *cis*-regulatory elements remain unknown. Furthermore, while it is generally accepted that many genes are controlled by multiple regulatory elements (*Barolo, 2012*; *Miguel-Escalada et al., 2015*) our understanding of multi-modular gene regulation remains limited, particularly in the context of mammalian development and stem cell differentiation.

Over the last decade, chromosome conformation capture (3C) and derived methods have enabled the biochemical mapping of looping interactions to offer new insights into their architecture across different cell types (*Dekker et al., 2013*; *de Laat and Duboule, 2013*; *Schmitt et al., 2016*). In particular, Hi-C has allowed genome-wide characterisation of higher order chromatin dynamics during differentiation at the level of contact domains, including A/B compartments and topologically associated domains (TADs) (*Fraser et al., 2015*; *Dixon et al., 2015*). The complexity of Hi-C samples creates challenges for a comprehensive identification of individual enhancer-promoter loops using this technology. However, analyses focusing on candidate-interacting regions or those bound by specific proteins (such as cohesin and RNA polymerase II) have made it possible to detect subsets of promoter-enhancer interactions at high resolution (*Heidari et al., 2014*; *Li et al., 2015*; *Sanyal et al., 2012*) and delineate their dynamics during cell differentiation and reprogramming. These studies provided evidence of interactions associated with transcriptional changes upon lineage commitment (*Denholtz et al., 2013*; *Kieffer-Kwon et al., 2013*; *Phillips-Cremins et al., 2013*;

*Zhang et al., 2013*), as well as revealed interactions formed in anticipation of changes in gene activity (*Apostolou et al., 2013*; *Ghavi-Helm et al., 2014*; *Wei et al., 2013*). However, despite these advances, the global and unbiased high-resolution mapping of promoter *cis*-regulatory interactions that form and remodel during development and stem cell differentiation is still lacking. This hampers an integrated understanding of the *cis*-regulatory logic underlying transcriptional decisions during lineage commitment.

Recently, we developed Promoter Capture Hi-C that uses sequence capture to enrich Hi-C libraries for interactions involving the promoters of most annotated genes, providing a global view on promoter interactions that is independent of the activity of interacting regions and identity of proteins recruited to them (*Mifsud et al., 2015*; *Schoenfelder et al., 2015a*). Here, we use PCHi-C in human embryonic stem cells (ESCs) and ESC-derived neuroectodermal cells (NECs) (*Bajpai et al., 2009*) to create a high-resolution resource of promoter contacts and their dynamics during early lineage commitment in the context of extensive chromatin changes that occur at the interacting *cis*-regulatory regions as the cells differentiate (*Rada-Iglesias et al., 2011*). Our large-scale dataset links thousands of known and predicted enhancer elements with their putative target genes in human pluripotent and early lineage-committed cells, including those known to drive tissue-restricted reporter gene expression in transgene assays. We integrate the promoter interacting regions of each gene to define *cis*-regulatory units (CRUs) that provide a view of multi-modular gene regulation. We show that CRUs undergo extensive reorganisation during lineage commitment that involves both the 'rewiring' (acquisition or loss) of promoter interactions, as well as chromatin state changes at pre-existing interactions. Importantly, we demonstrate that this reorganisation is associated with changes in target gene expression, thereby providing insights into the transcriptional control of early human development.

## Results

### A high-resolution atlas of promoter interactions in human pluripotent and early lineage-committed cells

We used PCHi-C to profile the interactomes of 21,841 promoters in human ESCs and NECs (*Figure 1A*). We generated NECs using an established protocol (*Rada-Iglesias et al., 2011*) (*Figure 1—figure supplement 1A*) and confirmed efficient differentiation by flow cytometry analysis and RNA-sequencing (*Figure 1—figure supplement 1B,C*). PCHi-C data normalisation and signal detection using the CHiCAGO pipeline (*Cairns et al., 2016*) identified 75,795 significant *cis*-interactions between promoters and other genomic regions in ESCs and 75,624 in NECs. In addition, approximately 300 significant *trans*-interactions were detected in each cell type. As examples of this rich dataset, high-confidence interactions are shown for the *SOX2* and *PAX6* promoters (*Figure 1B* and *Figure 1—figure supplement 2A*). These examples illustrate the multiple promoter-contacts observed, alongside the conventional Hi-C profiles additionally generated in this study that reveal higher-order genome topology over the same region. Overall, PCHi-C samples showed an 11 to 15-fold enrichment for promoter-containing interactions over conventional Hi-C. This data resource provides a global, high-resolution atlas of chromosomal interactions in human pluripotent and lineage-committed cells. Processed datasets have been made available through Open Science Framework (http://osf.io/sdbg4), and raw sequencing reads have been deposited to Gene Expression Omnibus (accession GSE86821).

### Identification of putative regulatory elements and their associated gene promoters

To gain insight into the chromatin properties of the promoter-containing interactions, we integrated the PCHi-C data with published genome-wide histone modification profiles in ESCs and NECs (*Rada-Iglesias et al., 2011*). In both cell types, promoter-interacting regions (PIRs) were significantly enriched for histone marks that are associated with regulatory functions (*Figure 1C*), including H3 lysine 4 monomethylation (H3K4me1) and H3 lysine 27 acetylation (H3K27ac), which can identify enhancers in human cell types, as well as H3 lysine four trimethylation (H3K4me3), which is associated with transcriptional activation, and H3 lysine 27 trimethylation (H3K27me3), which is associated with Polycomb-mediated transcriptional repression (*Di Croce and Helin, 2013*; *Heintzman et al.,*

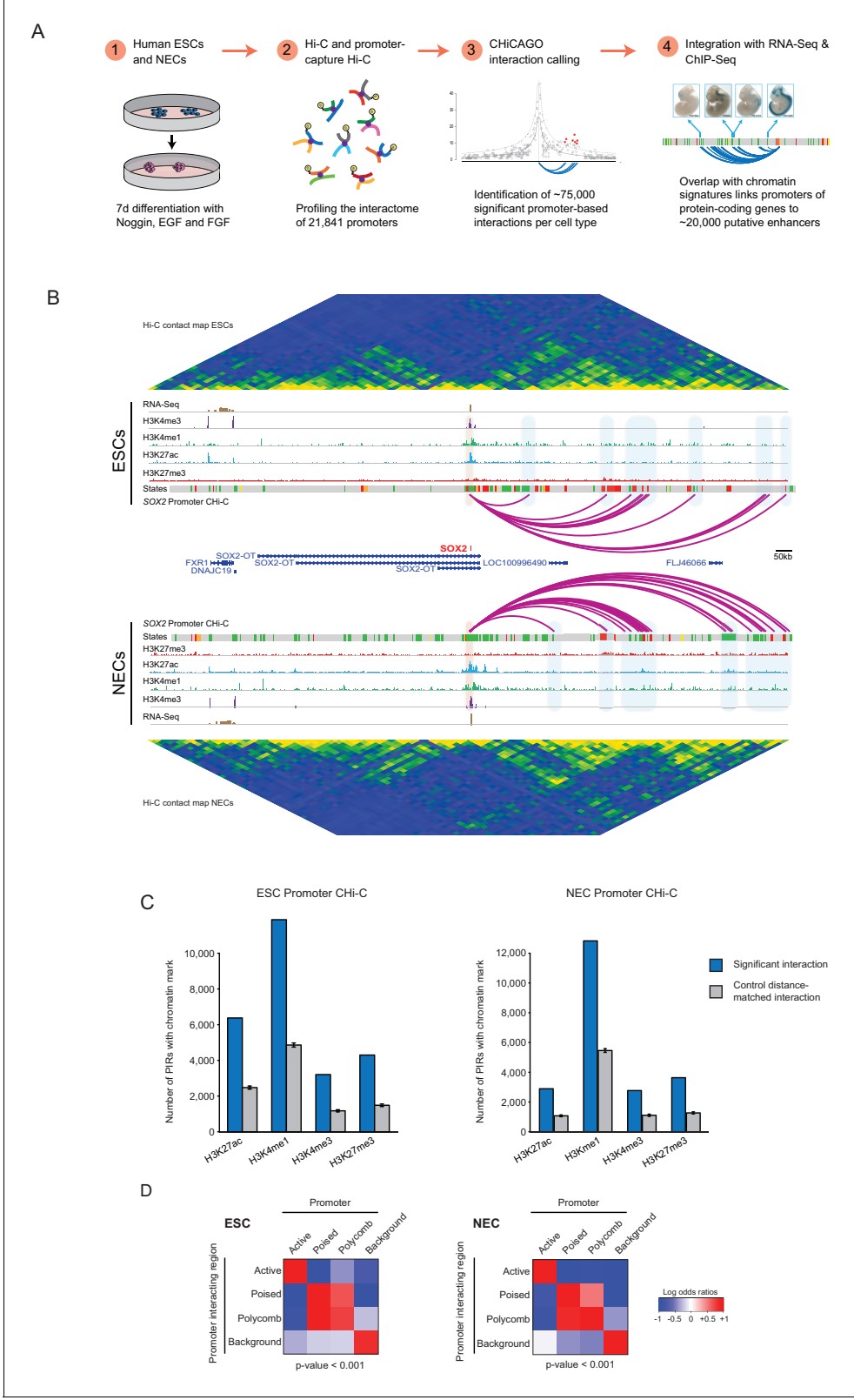

**Figure 1.** A resource of high-resolution promoter interactions in human embryonic stem cells (ESCs) and ESC-derived neuroectodermal cells (NECs). (**A**) Overview of the experimental design. Human embryonic stem cells (ESCs) and ESC-derived neuroectodermal progenitors (1) were analysed with Promoter Capture Hi-C to profile interactions involving 21,841 promoter-containing *HindIII* fragments (2). Signal detection with the CHiCAGO pipeline revealed ~75,000 high-confidence promoter interactions in each cell type (3). These data were integrated with histone modification and gene

*Figure 1 continued on next page*

*Figure 1 continued*

expression profiles in the same cells (4) to study chromatin and interaction dynamics during lineage commitment. Characterisation of ESCs and NECs is shown in *Figure 1—figure supplement 1*. (B) Genome browser representation of the *SOX2* promoter interactome in ESCs (upper) and NECs (lower). Significant interactions are shown as purple arcs, with one end of the interaction within the *SOX2* promoter and the other end at a promoter-interacting region (PIR). ChIP-seq (H3K27me3, H3K27ac, H3K4me1, H3K4me3; from [*Rada-Iglesias et al., 2011*]) and mRNA-seq tracks are shown. Chromatin states for each genomic region were defined by ChromHMM (*Ernst and Kellis, 2012*) using ChIP-seq data (active chromatin, green; poised chromatin, orange; Polycomb-associated chromatin, red; intermediate, yellow; background, grey). Conventional Hi-C heatmaps of contact frequencies reveal chromatin topology over this region. As an additional example, the *PAX6* promoter interactome is shown in *Figure 1—figure supplement 2*. Read count interaction profiles for *SOX2* and *PAX6* are shown in *Figure 1—figure supplement 4*. (C) PIRs are significantly enriched in regions that contain histone marks associated with putative regulatory functions, compared with promoter distance-matched control regions (permutation test p-value<0.01 for each mark) (ESCs, left; NECs, right). Blue bars show the number of overlaps observed in detected PIRs, and grey bars show the mean number of overlaps observed in distance-matched random regions over 100 permutations. Error bars show 95% confidence intervals across permutations. (D) Promoters and their associated PIRs show significant concordance in chromatin states. Heatmaps show the log2 odds ratios for the co-occurrence of each combination of promoter and PIR chromatin state compared with that expected at random. p-Values are from Pearson's $\chi^2$ test on the corresponding contingency tables. Clustering of chromatin states and additional examples of promoter interactomes are shown in *Figure 1—figure supplement 3*.

The following figure supplements are available for figure 1:

**Figure supplement 1.** Characterisation of ESCs and NECs.

**Figure supplement 2.** *PAX6* promoter interactome and CTCF enrichment at PIRs.

**Figure supplement 3.** Integrated view of chromatin states and PCHi–C data.

**Figure supplement 4.** Read-count interaction profiles for baited promoters presented in *Figures 1–3*.

*2007*, *2009*). PIRs were also significantly enriched for sites bound by the architectural protein CTCF (*Figure 1—figure supplement 2B*; based on ENCODE data available for ESCs only), consistent with previous observations in other cell types (*Jin et al., 2013*; *Phillips-Cremins et al., 2013*; *Sanyal et al., 2012*).

We used ChromHMM (*Ernst and Kellis, 2012*) to integrate these histone marks and to define four combinatorial chromatin states in both ESCs and NECs, as follows: active (characterised by H3K4me3 and/or H3K27ac); Polycomb-associated (H3K27me3); poised (H3K4 methylation and H3K27me3); and background (no detectable signal for the tested histone modifications) (*Figure 1—figure supplement 3A*; see Materials and methods for details). Overall, we detected just under 20,000 PIRs in each cell type that harboured either active (12,250 in ESCs and 7510 in NECs), Polycomb-associated (3505 in ESCs and 5856 in NECs) or poised (2274 in ESCs and 4262 in NECs) chromatin state signatures, connecting a large set of putative regulatory sequences in human pluripotent and lineage committed cells to their target promoters. In addition, 25,727 PIRs in ESCs and 20,016 PIRs in NECs were in the background state.

The chromatin states of several example promoters, including those for the *POU5F1*, *PRDM14* and *CHD7* genes, together with each of their respective PIRs, are shown in *Figure 1—figure supplement 3B*. When analysing the whole dataset, we found a significant concordance between the chromatin states at promoters and their PIRs (*Figure 1D*), which is in line with previous studies in other human cell types (*Jin et al., 2013*; *Mifsud et al., 2015*; *Sanyal et al., 2012*) and provides validation of our dataset. Notably, poised and Polycomb-associated promoters showed similar interaction preferences for PIRs in either of these two states (*Figure 1D*). This finding suggests that poised and Polycomb-associated regions are broadly interchangeable in terms of their interaction affinities, which is consistent with a key role for Polycomb-group proteins in mediating interactions in the poised state (*Schoenfelder et al., 2015b*).

Taken together, these data provide a comprehensive resource that links many thousands of known and predicted regulatory elements with their putative target genes and will enable the investigation of regulatory contacts during human lineage commitment.

## Promoter-interacting regions can function as tissue-restricted developmental enhancers

The enrichment of PIRs for specific chromatin regulatory features points to the presence of functional enhancer elements at these regions that could potentially provide inputs to the promoters they contact. To assess the enhancer activity of the identified PIRs, we examined whether they were known to efficiently drive reporter gene expression in embryonic day 11.5 mouse embryos based on information from the VISTA Enhancer Browser (*Visel et al., 2007*). As an initial example, we focused on the 39 PIRs detected in NECs that interact with the promoter of the neural transcription factor *POU3F2*. Strikingly, four out of the five *POU3F2* PIRs tested experimentally in VISTA transgenic assays showed reporter activity exclusively in neural tissues, and one PIR was inactive (*Figure 2A*). Furthermore, the mRNA expression pattern of mouse *Pou3f2* broadly matched the combined tissue-restricted pattern of the tested human *POU3F2* PIRs (*Figure 2A*). These results highlight how PCHi-C can contribute to our understanding of the *cis*-regulatory networks for key developmental genes.

We next examined all putative enhancer elements and their PCHi-C-identified promoter targets. Overall, 219 PIRs in ESCs and 267 PIRs in NECs overlapped VISTA-annotated human elements (*Supplementary file 2*). Notably, we found that NEC PIRs were strongly enriched for sequences that could drive reporter-gene activity in neural tissues and in other neuroectodermal derivatives, such as the neural tube and cranial structures (z-score = 11, *Figure 2B* and *Figure 2—figure supplement 1A*). In contrast, ESC PIRs were enriched for sequences active in neural (z-score = 6.7) and non-neural tissues (z-score = 4.5) at similar levels (*Figure 2B* and *Figure 2—figure supplement 1A*). Interestingly, the enrichment of PIRs with neural enhancer activity in NECs relative to their enrichment in ESCs was even more pronounced when we focused on PIRs in an active chromatin state (*Figure 2—figure supplement 1B*). Collectively, these results validate the function of several hundreds of PIRs as cell-type-specific developmental enhancers.

We next sought to link enhancers documented in the VISTA Enhancer Browser to their putative target genes on the basis of PCHi-C data. We detected the interactions of 267 VISTA-annotated human enhancers with 277 target gene promoters in NECs (*Supplementary file 2*). Of these, 122 PIRs (46%) interacted with their nearest gene, which is consistent with their current annotation in the VISTA Enhancer Browser. The remaining PIRs, however, did not interact with their nearest gene in NECs, but engaged with more distal promoters (*Supplementary file 2*). *Figure 2C* shows PCHi-C-based reassignment of enhancer targets for several examples of key neural regulators including *SOX2, SOX4,* and *FZD3* (*Figure 2C*), and the full results are listed in *Supplementary file 2*.

Taken together, these findings provide a functional validation of the detected human PIRs, and identify the putative promoter targets of multiple known enhancers.

## The *cis*-regulatory unit: an integrated view of promoter interactions

We found interacting promoters to engage a median of four PIRs (*Figure 3A*), consistent with findings in other human cell types (*Jin et al., 2013*; *Sanyal et al., 2012*). To obtain an integrated view of promoter interactions, we considered PIRs connected to each promoter to jointly form a '*cis*-regulatory unit' (CRU, *Figure 3B*). Focusing on protein-coding genes, and considering all promoters associated with at least one PIR, we defined 9008 CRUs in ESCs and 9361 in NECs, and studied their localisation, chromatin properties and dynamics during cell lineage commitment.

CRUs spanned a median of ~230 kb (with a range of 1 kb-200Mb) in both cell types (*Figure 3C*). We assessed their localisation with respect to higher order features of chromosome architecture including TADs and Insulated Neighborhoods (INs) (*Dixon et al., 2012*; *Ji et al., 2016*; *Nora et al., 2012*; *Sexton et al., 2012*). We defined TADs in ESCs and NECs using Hi-C data for these cells generated as part of this study (see Materials and methods). Overall, ~75% of CRUs were fully contained within a TAD in ESCs and NECs, which was significantly higher than expected by random (*Figure 3D–G* and *Figure 3—figure supplement 1D,E*). In the remaining ~25% of CRUs, either some or all PIRs localised outside of the promoter-harbouring TAD (*Figure 3H* and *Figure 3—figure supplement 1D*). We found that TAD boundaries crossed by promoter interactions were generally weaker than non-crossed boundaries (Wilcoxon test p-value=1.8e-14; *Figure 3—figure supplement 1H*). However, the ranges of strength scores for 'crossed' and 'non-crossed' TAD boundaries were

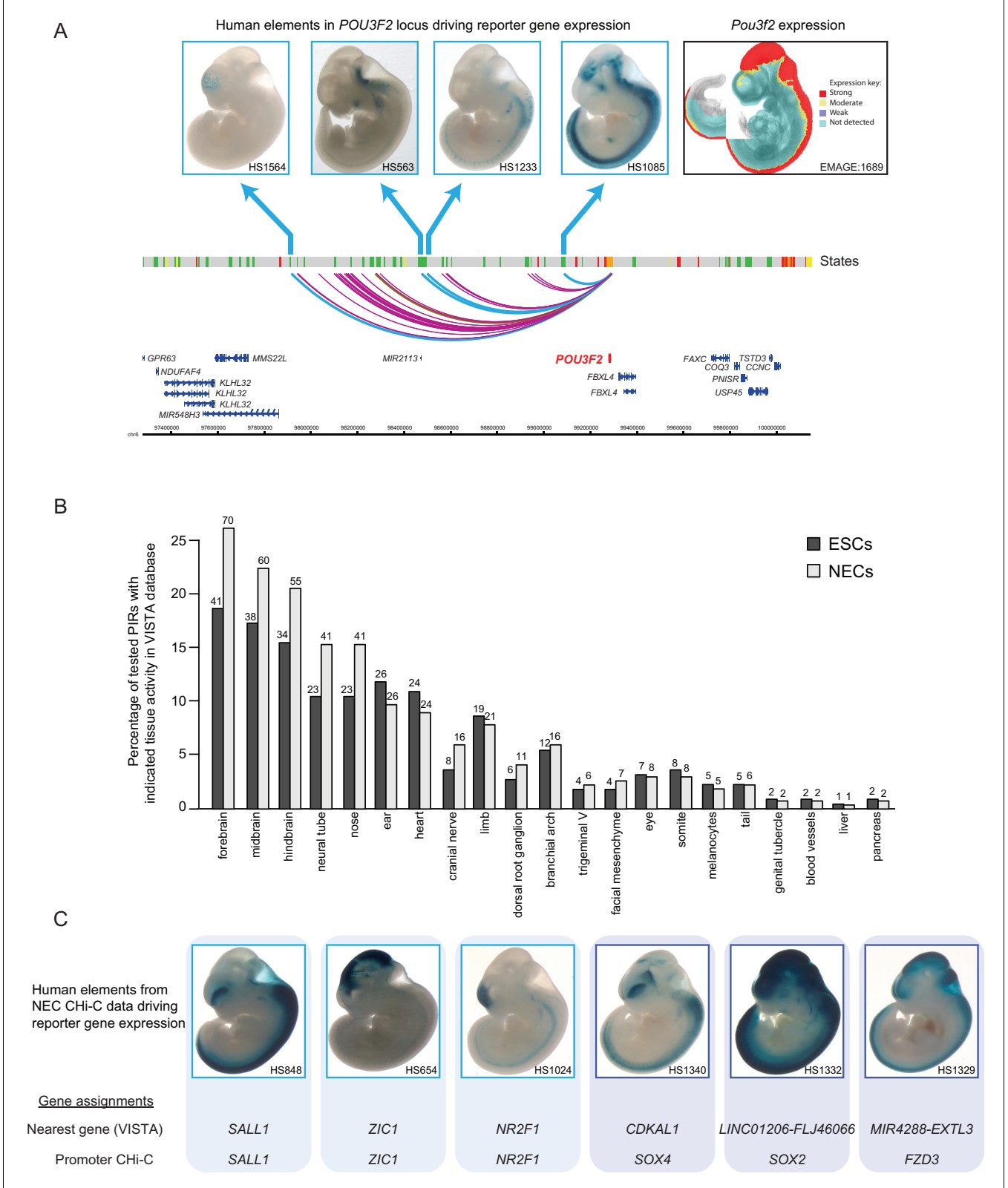

**Figure 2.** Promoter-interacting regions can function as tissue-restricted developmental enhancers and identify associated target genes. (**A**) A genome browser representation of the *POU3F2* promoter interactome in NECs. Genome coordinates are shown underneath. Chromatin states are indicated (active chromatin, green; poised chromatin, orange; Polycomb-associated chromatin, red; intermediate, yellow; background, grey). Significant interactions are shown as coloured arcs. Five of the identified *POU3F2* PIRs have been tested experimentally using a transgenic reporter assay as part

*Figure 2 continued on next page*

*Figure 2 continued*

of the VISTA Enhancer Browser (*Visel et al., 2007*). Of those five, four regions (indicated by blue arcs) can drive tissue-restricted *LacZ* expression in E11.5 mouse embryos. Representative images of X-gal stained mouse embryos are shown for each sequence. These show neural-restricted enhancer activity within the forebrain, midbrain, hindbrain and neural tube, which are tissues derived from NECs. The mRNA expression pattern of *Pou3f2* in an E10.5 mouse embryo (EMAGE gene expression database; EMAGE:1689; [*Richardson et al., 2014*]) broadly matches the combined tissue-restricted pattern of its enhancers. One experimentally tested PIR (indicated by brown arc) is inactive at this developmental stage in mouse embryos. (**B**) PIRs identified in NECs are enriched for sequences that can drive reporter gene activity in neural tissues and other neuroectodermal derivatives (see also *Figure 2—figure supplement 1A*). The barplot shows the distribution of tissue-specific reporter expression patterns for all experimentally tested PIRs identified in ESCs (n = 219) and NECs (n = 267). Embryo reporter assays and enhancer activity patterns are from the VISTA Enhancer Browser (*Visel et al., 2007*). The number of PIRs active within a particular tissue is shown above each bar. PIRs with an active chromatin state in NECs showed an even more pronounced enrichment for enhancer activity in neural tissues (*Figure 2—figure supplement 1B*). (**C**) Representative images of X-gal stained mouse embryos from the VISTA Enhancer Browser (*Visel et al., 2007*) reveal neural-restricted reporter gene activity for six example NEC PIRs. Shown underneath is the gene promoter assignment for the associated enhancer in VISTA and in our PCHi-C dataset.

The following figure supplement is available for figure 2:

**Figure supplement 1.** Active PIRs are enriched for enhancers with neural-specific activity.

highly overlapping, and even some of the strongest boundaries were penetrable to interactions (*Figure 3—figure supplement 1H*).

For INs, we used the published genomic coordinates (available for ESCs only) that were defined on the basis of cohesin ChIA-PET and CTCF-binding data (*Ji et al., 2016*). Just under 30% of CRUs were fully contained within IN boundaries, and this proportion increased to ~45% when considering the largest span of each overlapping set of INs as a single unit (*Figure 3I* and *Figure 3—figure supplement 1I*). These numbers significantly exceeded the proportions expected at random (*Figure 3I* and *Figure 3—figure supplement 1I*), but at the same time, also provided abundant examples of IN-spanning CRUs (*Figure 3G,H*). Taken together, these results suggest that CRUs are partially constrained by, but not fully contained within, higher order topological structures such as TADs and INs.

## ESC differentiation is associated with the dynamic reorganisation of CRUs

To investigate the potential regulatory features of CRUs, we first characterised their chromatin properties by considering the proportion of PIRs in each chromatin state within a CRU. Applying hierarchical clustering based on this property, we obtained eight distinct clusters of CRUs in both ESCs (*Figure 4A*) and NECs (*Figure 4—figure supplement 1*), corresponding to different combinations of PIR chromatin states within CRUs. We found that CRUs within three prevalent clusters contained PIRs in one predominant, non-background, chromatin state (clusters 1–3; *Figure 4A* and *Figure 4—figure supplement 1A*). In contrast to these 'uniform' CRUs, 18% of CRUs in ESCs and 24% in NECs contained combinations of PIRs in active, poised and Polycomb-associated chromatin states (clusters 4–7; *Figure 4A* and *Figure 4—figure supplement 1*). Finally, CRUs in cluster 8 contained PIRs exclusively in the background state (*Figure 4A* and *Figure 4–figure supplement 1*). Examples of genes in ESCs assigned to the different CRU clusters are shown in *Figure 4B* and *Figure 4—figure supplement 2*. Notably, the chromatin state of each promoter generally matched that of the most prevalent CRU chromatin state (*Figure 4A* and *Figure 4—figure supplement 1*). Overall, this classification provides a framework for exploring CRU properties.

We set out to investigate CRU chromatin state transitions on ESC to NEC differentiation. For this analysis, and in each cell type, we classified CRUs into either single-state active (containing active, and possibly also background-state, PIRs), single-state repressed (containing poised and/or Polycomb-associated, and possibly also background-state PIRs), or background (containing only background-state PIRs). CRUs containing a combination of both active and repressed (Polycomb-associated/poised) PIRs were classified as dual-state. We found that 65% of the single-state CRUs in ESCs remained single-state CRUs in NECs, although approximately half of them switched their state (*Figure 5A*). In addition, similar proportions of CRUs lost (11%) and acquired (13%) a dual-state configuration on ESC to NEC differentiation. These findings demonstrate that considerable reorganisation of CRUs occurs during lineage commitment.

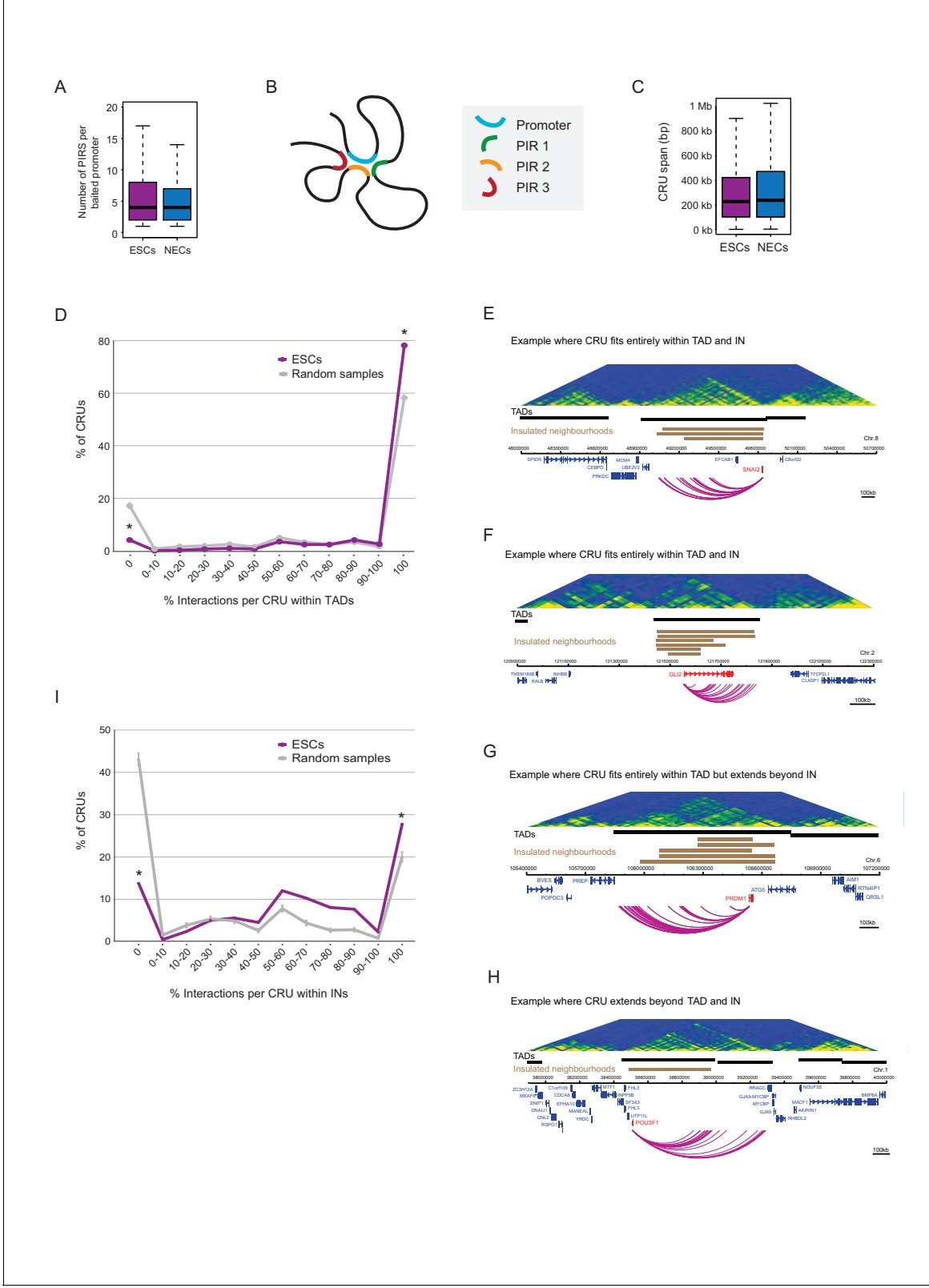

**Figure 3.** Characterisation of *cis*-regulatory units (CRUs). (**A**) Boxplot shows the distributions of the number of PIRs per interacting promoter in ESCs (n = 17955) and NECs (n = 18146). Promoters with no detected PIRs are not shown (4121 in ESCs; 3930 in NECs). The number of interactions per promoter showed only a minor dependence on transcriptional activity and promoter chromatin state (*Figure 3—figure supplement 1A,B*). (**B**) A schematic illustrating the concept of a CRU as a collection of all PIRs together with their associated promoter. Note that it cannot be ruled out that some PIRs

*Figure 3 continued on next page*

*Figure 3 continued*

may provide alternative rather than concurrent interactions. (C) Boxplot shows the distributions of CRU span in ESCs and NECs. We observed a moderate dependence between the span and the number of PIRs (*Figure 3—figure supplement 1C*). (D) CRUs are preferentially contained within an individual TAD. Line graph shows the percentage of CRUs with different proportions of interactions that reside within an individual TAD (purple) and the summary statistics (mean and 95% confidence error bars) obtained for 1000 random samples, keeping the same CRU structure (grey). There is a significant tendency for CRUs to be contained entirely within a TAD ( * denotes permutation test p-value<0.001). In addition, fewer CRUs span entirely over a TAD boundary (* denotes permutation test p-value<0.001). The 1000 random samples were generated by permutations of CRUs across all promoter fragments, retaining the same overall CRU structure. Error bars show 95% confidence intervals. Data shown are for ESCs (n = 9008 CRUs); data for NECs are shown in *Figure 3—figure supplement 1D*. We found that CRUs crossing TAD and IN boundaries generally contained a higher number of PIRs (*Figure 3—figure supplement 1F,G*). E–H) Genome browser representations of CRUs in ESCs. Examples include the *SNAI2* CRU (E) and *GLI2* CRU (F), which both fit entirely within a TAD and INs; *PRDM1* CRU (G), which fits entirely within a TAD but extends beyond INs, and *POU3F1* CRU (H), which extends over a TAD boundary and also beyond an IN. (I) CRUs are preferentially contained within INs, but interactions can extend beyond IN boundaries. The line graph shows the percentage of CRUs with different proportions of interactions that reside within an individual IN in ESCs (coordinates obtained from [*Ji et al., 2016*]). There is a significant tendency for CRUs (purple) to be contained entirely within an IN, compared to random (grey) (* denotes p-value<0.001 from a permutation test done with 1000 random samples). In addition, fewer CRUs span entirely beyond an IN (* denotes p-value<0.001 from a permutation test done with 1000 random samples). Error bars show 95% confidence intervals. Promoters outside of a defined IN were excluded from the analysis.

The following figure supplement is available for figure 3:

**Figure supplement 1.** Additional CRU characterisation.

CRU state transitions associated significantly with changes in the expression of their target genes (p-value<0.005, Fisher's exact test; *Figure 5B,C*). In particular, genes that were transcriptionally upregulated upon ESC differentiation preferentially gained an active single-state in NECs, either through switching the chromatin state of a single-state CRU (*Figure 5B*) or through resolving a dual-state CRU (*Figure 5C*). Examples of CRUs undergoing each scenario include *RGMB* and *MAP2*, which are transcriptionally upregulated in NECs (*Figure 5D,E*). Pronounced chromatin changes were also detected at the CRUs of genes downregulated upon differentiation, including a loss of the active single-state and/or a transition to the repressed single-state ( *Figure 5B,C*; example shown in *Figure 5F*). Taken together, these results suggest that the modulation of CRU chromatin state is associated with transcriptional changes upon ESC differentiation. This modulation might potentially underlie many transcriptional changes in early lineage commitment.

## Rewiring and recolouring of promoter-interacting regions contribute jointly to gene expression dynamics upon lineage commitment

To investigate the underlying processes that drive changes in CRU organisation during cell lineage commitment, we studied the dynamics of promoter interactions and chromatin states at the individual PIRs. We refer to changes in PIR connectivity as 'rewiring', and to chromatin state changes at PIRs as 'recolouring', and note that they do not need to be mutually exclusive (*Figure 6A*). To distinguish between interactions that are rewired and retained on ESC differentiation at high confidence, we applied additional filters to the PCHi-C data, resulting in 1153 rewired (present in only one cell type) and 1258 retained (present in both ESCs and NECs) interactions (see Materials and methods for details).

Importantly, we found that the co-occurrence of rewiring and recolouring interactions on ESC differentiation was significantly more common than expected at random (*Figure 6B*, p-value<0.001, Fisher's exact test). Specifically, new interactions that were gained by NECs preferentially acquired the active state, or transitioned from the background to repressed state (*Figure 6B*, bottom row). Interactions that were lost on ESC differentiation were enriched for PIRs that transitioned from the active to poised/Polycomb-associated states, as well as for those switching to the background state (*Figure 6B*, middle row). Notably, the vast majority of rewiring events (99.7%) were not associated with larger-scale A/B compartment dynamics (not shown). Together, these observations indicate that lineage commitment associates with concerted changes in the connectivity and chromatin state of regulatory regions.

Interactions at *NR2F1* (*Figure 6C*) and *ZSCAN2* (*Figure 6—figure supplement 1A*) exemplify the preferential co-occurrence of rewiring and recolouring events, with interactions present in the cell

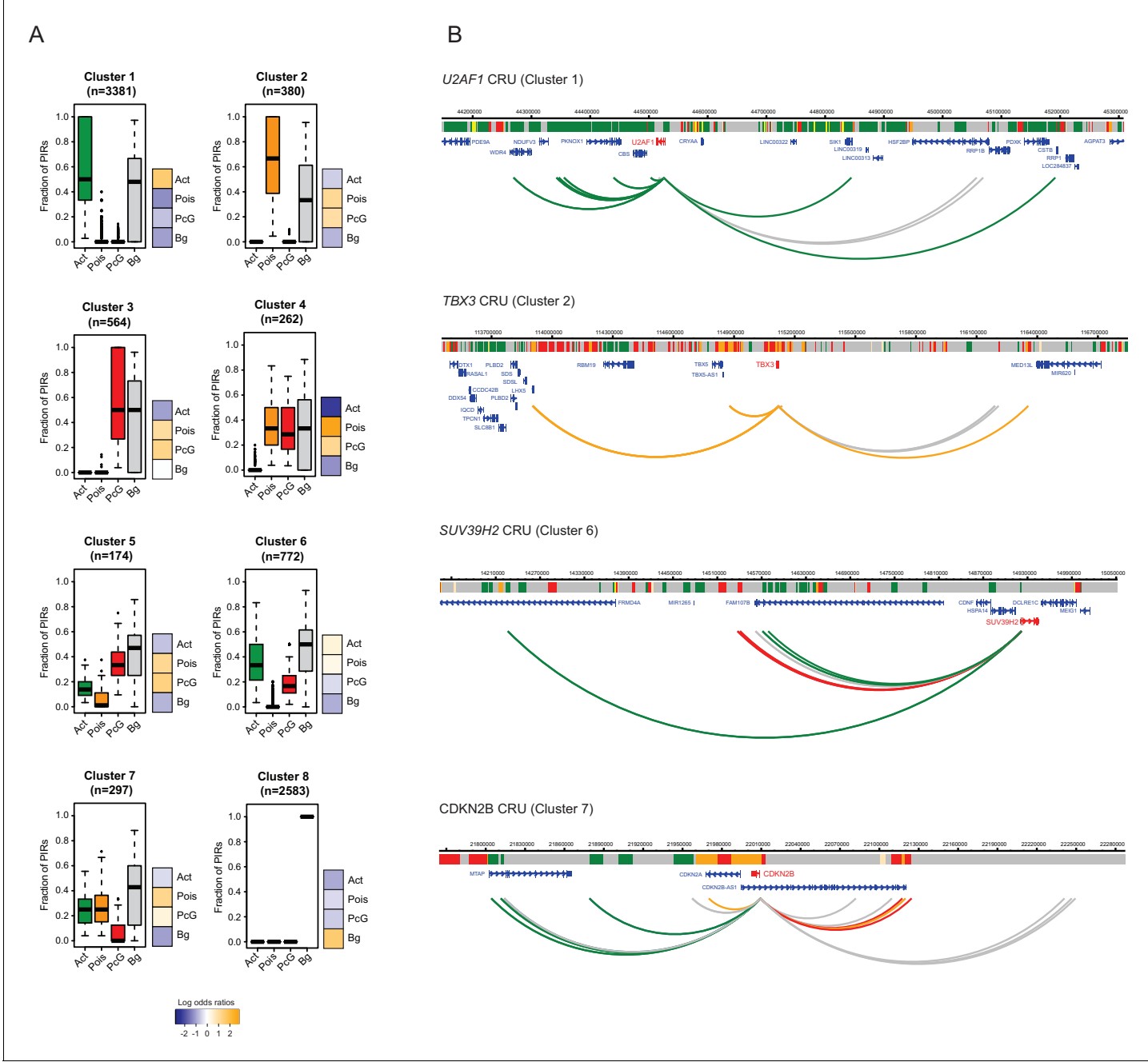

**Figure 4.** Clustering of CRUs according to chromatin state of each PIR in ESCs. (**A**) CRUs from ESCs were clustered hierarchically according to the distribution and fractions of their PIRs that correspond to each chromatin state. Boxplots show the distribution of PIR fractions for each chromatin state (Act, active; Pois, poised; PcG, Polycomb-associated; Bg, background). Heatmaps show the log2 odds ratios of observing each promoter state associated with a CRU in each cluster (p<0.001, $\chi^2$ test on the contingency table). Data for NECs are shown in *Figure 4—figure supplement 1*. (**B**) Genome browser representations of CRUs in ESCs. *U2AF1* CRU from cluster 1 and *TBX3* CRU from cluster 2, each exemplify cases where non-background PIRs within a CRU are associated with a uniform chromatin state. *SUV39H2* CRU from cluster 6 and *CDKN2B* CRU from cluster 7, each exemplify cases where PIRs within a CRU are associated with multiple chromatin states. Interaction arcs are coloured according to PIR chromatin state (active, green; poised, orange; Polycomb-associated, red; background, grey). See *Figure 4—figure supplement 2* for additional examples, and *Figure 4—figure supplement 3* for read count interaction profiles.

The following figure supplements are available for figure 4:

**Figure supplement 1.** Clustering of CRUs according to chromatin state of each PIR in NECs.

*Figure 4 continued on next page*

*Figure 4 continued*

**Figure supplement 2.** Additional examples of CRUs in ESCs.

**Figure supplement 3.** Read-count interaction profiles for baited promoters presented in *Figures 4* and *5*.

type in which the respective PIR is in the active state. However, we also found examples of PIR rewiring that showed unchanged chromatin states in both cell types, such as those at the *JAG1* and *HAPL3* genes (*Figure 6D* and *Figure 6—figure supplement 1B*). Finally, we observed that 25% of PIRs that were retained in both cell types undergo chromatin state recolouring (*Figure 6E*, red segment). These regions included, for example, PIRs associated with the *IRX3* and *RAB3B* promoters (*Figure 6F* and *Figure 6—figure supplement 1C*).

We asked how rewiring and recolouring events at PIRs contribute to gene expression dynamics on ESC to NEC differentiation. We found that the loss or gain of interactions with active-state PIRs associated significantly with changes in gene expression (*Figure 6G*), suggesting their functional contribution to transcriptional control. Notably, gene expression changes were detected at retained and recoloured PIRs (*Figure 6G*, left panel), and also when active-state PIRs were lost or gained through rewiring (*Figure 6G*, right panel).

Taken together, these findings demonstrate that chromatin state changes and rewiring of interactions at PIRs contribute jointly to transcriptional regulation. Furthermore, our results show that promoter interaction dynamics preferentially co-occur with chromatin state dynamics upon cell lineage commitment.

## Discussion

### A comprehensive atlas of promoter interactions

Our study presents an atlas of promoter interactions in human pluripotent and early lineage-committed cells, and offers new insights into the association between genome organisation and gene regulation. The high resolution of PCHi-C has enabled us to detect individual promoter-associated loops at a single restriction enzyme fragment resolution. We find that promoter-interacting regions in both cell types harbour multitudes of previously known and putative enhancer elements, which we link with their physically associated target genes. While the identified connections are predictive of regulatory relationships, it is important to note that the current data are correlative and will require functional validation using targeted genetic approaches and reporter assays. In addition to active enhancers, we find extensive promoter connectivity to regions associated with Polycomb-mediated repression and poising, reinforcing the role of Polycomb-group proteins in controlling chromosomal topology at transcriptionally inactive genes (*Entrevan et al., 2016*; *Li et al., 2015*; *Schoenfelder et al., 2015b*; *Vieux-Rochas et al., 2015*). Consistent with previous observations (*Sanyal et al., 2012*), we also detect large numbers of interactions between promoters of both active and inactive genes, and regions devoid of chromatin features. It is possible that such interactions are structural, rather than play gene regulatory roles. However, a regulatory function for some 'unmarked' PIRs also cannot be ruled out as recent mutagenesis experiments have identified functional elements that lack 'classic' chromatin annotations (*Pradeepa et al., 2016*; *Rajagopal et al., 2016*).

### Reconfiguration of *cis*-regulatory units upon lineage commitment

The high-resolution promoter-interaction information has enabled the identification of *cis*-regulatory units (CRUs) as sets of interactions connected to the same promoter. Taking the view of CRUs, we consider jointly the dynamics of chromatin states and interactions as ESCs differentiate, and assess the potential contribution of these processes to changes in gene expression during development. We observe that CRUs reconfigure extensively upon cell differentiation. These include the CRUs of ESC- and NEC-specific genes, for which CRU reconfiguration associates with transcriptional changes upon differentiation, as well as the CRUs of genes that are not expressed in either cell type,

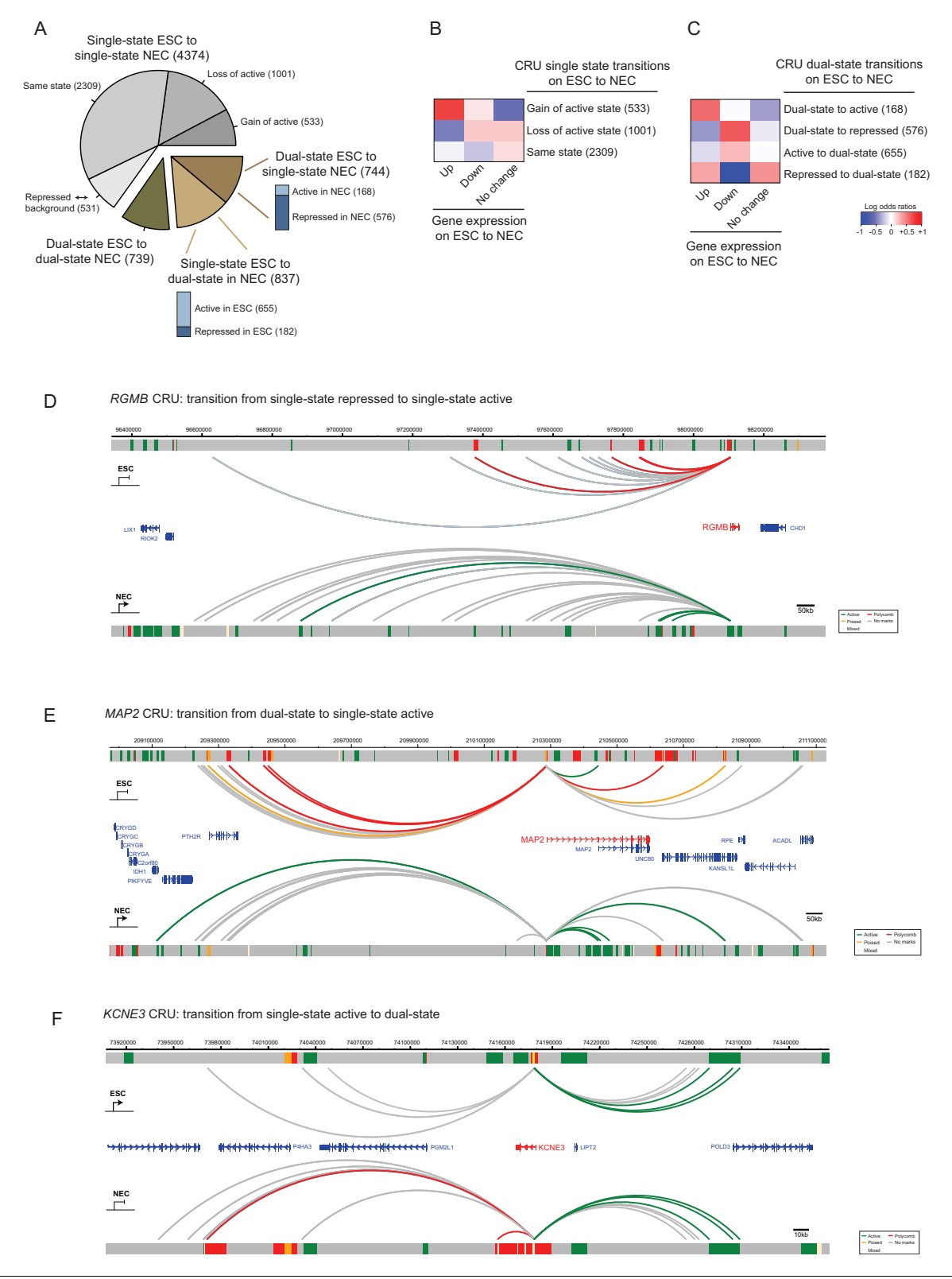

**Figure 5.** CRU state transitions occur during ESC differentiation and are associated with changes in gene transcription. (A) Pie chart summarising CRU state transitions that occur upon ESC to NEC differentiation. The number of CRUs within each transition category are shown. Transitions that involve dual-state to single-state, and single-state to dual-state, are further subdivided into whether the single-state is classified as active or repressed (Polycomb-associated or poised). (B–C) Heatmaps show the log2 odds ratios for CRU state transitions and associated changes in gene expression. (B)

*Figure 5 continued on next page*

Figure 5 continued

Single-state transitions showing a non-random segregation withgene expression changes (p-value=0.0031, Fisher's exact test); (**C**) dual-state transitions showing a non-random segregation with gene expression changes (p-value=0.0014, Fisher's exact test). Number of CRUs within each transition category are shown. Genes differentially expressed between ESCs and NECs were identified using DESeq2 (FDR < 0.05 and a log2 fold change of >1.5). Repressed state includes Polycomb-associated and poised states. (**D–F**) Genome browser representations of CRU state transitions that occur upon the differentiation of ESCs (top image) to NECs (lower image). (**D**) *RGMB* provides an example of a CRU transitioning from a single repressed to a single active state and an associated increase in *RGMB* transcription. (**E**) *MAP2* provides an example of a dual-state to a single active state CRU transition and an associated increase in *MAP2* transcription. (**F**) *KCNE3* provides an example of a single active state to a dual-state CRU transition and an associated decrease in *KCNE3* transcription. Arcs are coloured according to PIR chromatin state (active, green; poised, orange; Polycomb-associated, red; background, grey).

consistent with the model of progressive chromatin changes at lineage-inappropriate genes during lineage commitment (*Spivakov and Fisher, 2007*).

Previous studies on candidate loci have proposed that cell-state changes are associated with two predominant models of enhancer–promoter interaction dynamic that have been termed instructive and permissive (*de Laat and Duboule, 2013*). Instructive interactions are established de novo upon cell state change and are concomitant with target gene activity. In contrast, permissive interactions are already in place before the gene activation occurs and may therefore contribute to enhancer priming. Examples of instructive and permissive interactions have been described in pluripotent cell reprogramming and differentiation (*Apostolou et al., 2013*; *Denholtz et al., 2013*; *Phillips-Cremins et al., 2013*; *Wei et al., 2013*; *Zhang et al., 2013*), but little was known on a global scale about which model of enhancer-promoter interactions is predominant during lineage commitment. Here, we show that developmental changes at CRUs involve both the rewiring of 'instructive' interactions and the recolouring of the chromatin state of 'permissive' interacting regions. Notably, we find that these two processes tend to occur hand in hand, with the strongest association occurring between cell-type-specific promoter interactions and the active state of the respective PIRs. Importantly, CRU chromatin dynamics (at both rewired and preformed interactions) associates with consistent changes in gene expression, suggesting that both mechanisms are functionally important in mediating lineage-specific transcriptional programmes. The exact determinants of 'permissive' versus 'instructive' interactions remain to be elucidated and may depend on the identity of *cis*-acting factors recruited to the regulatory regions, as well as on local chromatin environments.

## Implications for developmental gene regulation by multiple enhancers

The CRU view provides an opportunity to consider multi-modular gene regulation in early human development that has hitherto been studied on a limited number of genes, predominantly in model organisms (*Barolo, 2012*; *Cannavò et al., 2016*; *Hong et al., 2008*). The 'single-state' architecture that we detect at the majority of CRUs is in line with observations of 'shadow enhancers' with overlapping activities in *Drosophila* (*Hong et al., 2008*). It has been suggested that this *cis*-regulatory organisation ensures the robustness of gene regulation and can buffer the effects of deleterious sequence variation, as well as providing opportunities for evolutionary innovation (*Barolo, 2012*; *Cannavò et al., 2016*; *Hong et al., 2008*; *Perry et al., 2010*).

'Dual-state' CRUs, although representing a relative minority of the CRUs we analysed, offer additional insights into signal integration at promoters. Specifically, the fact that the chromatin state of the promoter largely associates with the predominant chromatin state of the connected PIRs suggests that promoters may integrate signals from remote elements based on 'majority vote logic'. Mechanistically, this logic may be a consequence of largely independent enhancer action (potentially at both single-state and dual-state CRUs) that is consistent with the 'hit-and-run' model of transcriptional regulation (*Schaffner, 1988*; *Varala et al., 2015*), and provides a flexible way to fine-tune the expression of multi-enhancer genes (*Guerrero et al., 2010*; *Lagha et al., 2012*). However, this model also does not preclude the possibility that promoter chromatin states at 'dual-state' CRUs undergo a continuous turnover depending on the state of the PIR they contact. In this case, the observed 'majority-vote' promoter chromatin states would correspond to the predominant state detected at the population level.

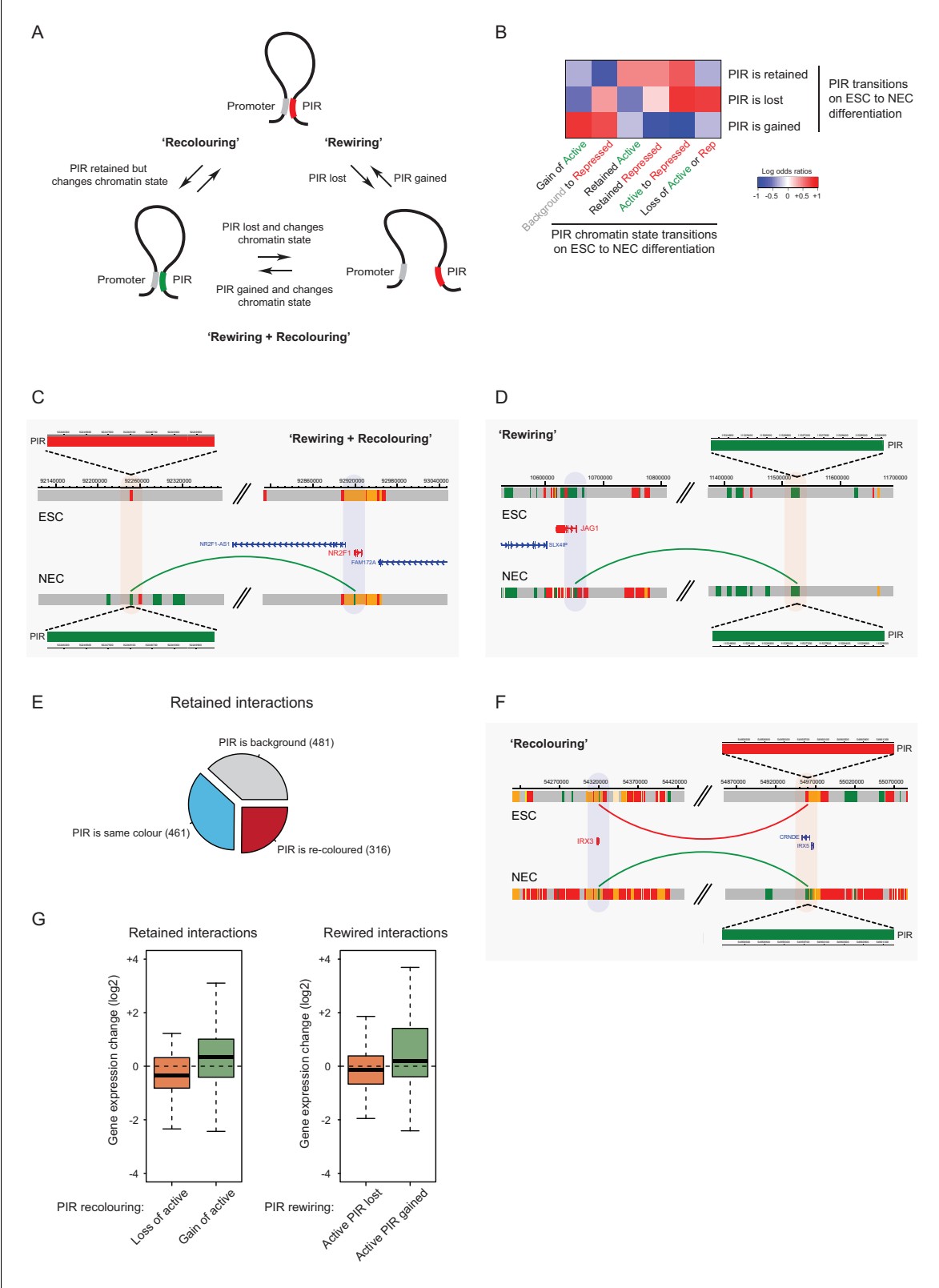

**Figure 6.** Interaction Dynamics: 'recolouring' versus 'rewiring'. (**A**) A schematic of interaction dynamics during cell differentiation. In a 'recolouring' interaction (left)the PIR undergoes a change in chromatin colour (reflecting a change in chromatin state) between the two cell types. In a 'rewiring' interaction (right), an interaction is gained or lost upon cell differentiation. In a 'rewiring + recolouring' interaction, the loss or gain of an interaction is concomitant with changes in chromatin colour at the respective PIR. (**B**) Heatmap of log2 odds ratios showing the association between different PIR

*Figure 6 continued on next page*

*Figure 6 continued*

chromatin state transitions (recolouring) and PIR interaction dynamics (rewiring) on ESC to NEC differentiation (p-value<0.001, Fisher's exact test). (**C–D**) Genome browser representations of interaction dynamics upon ESC to NEC differentiation. Note that only one interaction is shown for each example. Arcs are coloured according to PIR chromatin state (active, green; Polycomb-associated, red). (**C**) Rewiring and recolouring: upon differentiation, *NR2F1* gains an interaction with a PIR that is active in NECs, but repressed in ESCs. (**D**) Rewiring: the *JAG1* promoter gains an interaction with an active PIR in NECs. Additional examples are shown in *Figure 6—figure supplement 1A,B*. (**E**) Pie chart summarising the different scenarios in which an interaction is retained upon ESC to NEC differentiation. (**F**) Recolouring: the *IRX3* promoter retains an interaction, but the PIR changes from repressed (in ESCs) to active (in NECs). (**G**) Boxplots revealing the transcriptional changes as a function of active PIR dynamics during recolouring (left) and rewiring (right) events. In either scenario, there was a significant association between the acquisition and loss of an active state and changes in gene expression (p-values<0.001 for both recolouring and rewiring according to one-sided Wilcoxon rank sum tests).

The following figure supplement is available for figure 6:

**Figure supplement 1.** Interaction dynamics involving recolouring and rewiring.

Theoretically, the generally independent enhancer action also enables activation signals from individual elements to quantitatively 'add up' (at least to some extent) to promote stronger transcriptional outputs (*Arnold et al., 2013*; *Bothma et al., 2015*; *Lam et al., 2015*; *Spivakov, 2014*). Our observation that the resolution of dual-state CRUs toward a uniformly active state generally results in increased expression (and vice versa) supports this model. However, analyses in *Drosophila* have identified exceptions to additive enhancer activity (*Bothma et al., 2015*) and have provided examples of enhancers that activate more than one promoter in a coordinated fashion (*Fukaya et al., 2016*), which is not immediately expected from the 'hit-and-run' looping model. Finally, there is also a possibility that multiple enhancers are jointly engaged in 'chromatin hubs' with promoters, rather than acting individually (*Hanscombe et al., 1991*; *Jiang et al., 2016*; *Patrinos et al., 2004*; *Tolhuis et al., 2002*; *Wijgerde et al., 1995*). These mechanistic questions go beyond the capabilities of Hi-C-based analyses of cell populations, and as such it is possible that multiple promoter interactions detected within a CRU take place either concurrently or simultaneously. The emerging studies at the single-molecule level (such as [*Bartman et al., 2016*; *Fukaya et al., 2016*]) will likely shed further light on the molecular mechanisms that underpin the principles of CRU organisation.

Promoter – enhancer interactions are vitally important for gene regulation and their disruption may lead to pronounced developmental abnormalities (*Epstein, 2009*). The high-resolution resource of the promoter-interaction landscape in pluripotent and early lineage-committed cells presented here, therefore, provides a stepping stone to understanding the logic of gene regulation and its aberrations during human embryogenesis.

## Materials and methods

### Cell culture

ESCs (H9/WA09; obtained from WiCell (Madison, WI); RRID:CVCL_9773) were cultured at 37°C in 5% CO2 in air in Pluripro media and matrix (Cell Guidance Systems (Cambridge, UK)). Authentication of ESCs was achieved by confirming the expression of pluripotency genes and protein markers, and by SNP analysis of sequencing data. ESCs were routinely verified as mycoplasma-free using a PCR-based assay (Sigma (St. Louis, MO)). The H9/WA09 line is not on the list of commonly misidentified cell lines (International Cell Line Authentication Committee). ESCs were differentiated into NECs using a previously described protocol (*Rada-Iglesias et al., 2011*) and samples were harvested on day 7.

### Flow cytometry

Following dissociation with accutase, ESCs and NECs were stained on ice for 45 min with CD326-AF647 (BioLegend (London, UK), Cat# 324212, RRID:AB_756086; 5 µL per million cells) and CD56-PE (BD Biosciences (San Jose, CA), Cat# 345812, RRID:AB_2629216; 20 µL per million cells) antibodies in 100 µl PBS containing 2% FBS. After washing, DAPI was included at a final concentration of 5

µl/mL for live/dead cell discrimination, and flow cytometry analysis was performed using a BD LSRFortessa with subsequent data analysis using FlowJo V10.1.

## Hi-C and promoter capture Hi-C (CHi-C)

Hi-C and Promoter CHi-C libraries were generated essentially as described (*Mifsud et al., 2015*; *Schoenfelder et al., 2015a*), with minor modifications. 3 to $4 \times 10^7$ cells (ESCs or NECs) were fixed in 2% formaldehyde (Agar Scientific (Stansted, UK)) for 10 min, after which the reaction was quenched with ice-cold glycine (0.125 M final concentration). Cells were collected by centrifugation (400 x g for 10 min at 4°C), and washed once with PBS (50 ml). After another centrifugation step (400 x g for 10 min at 4°C), the supernatant was completely removed and the cell pellets were immediately frozen in liquid nitrogen and stored at −80°C. After thawing, the cell pellets were incubated in 50 ml ice-cold lysis buffer (10 mM Tris-HCl pH 8, 10 mM NaCl, 0.2% Igepal CA-630, protease inhibitor cocktail (Roche (Basel, Switzerland)) for 30 min on ice. After centrifugation to pellet the cell nuclei (650 x g for 5 min at 4°C), nuclei were washed once with 1.25 x NEBuffer 2. The nuclei were then resuspended in 1.25 x NEBuffer 2, SDS was added (0.3% final concentration) and the nuclei were incubated at 37°C for 1 hr with agitation (950 rpm). Triton X-100 was added to a final concentration of 1.7% and the nuclei were incubated at 37°C for 1 hr with agitation (950 rpm). Restriction digest was performed overnight at 37°C with agitation (950 rpm) with *HindIII* (NEB; 1500 units per 7 million cells). Using biotin-14-dATP (Life Technologies (Carlsbad, CA)), dCTP, dGTP and dTTP (all at a final concentration of 30 µM), the *HindIII* restriction sites were then filled in with Klenow (NEB (Ipswich, MA)) for 75 min at 37°C. After addition of SDS (1.42% final concentration) and incubation at 65°C with agitation (950 rpm) for 20 min, ligation was performed for 4 hr at 16°C (50 units T4 DNA ligase (Life Technologies) per 7 million cells starting material) in a total volume of 8.2 ml ligation buffer (50 mM Tris-HCl, 10 mM MgCl₂, 1 mM ATP, 10 mM DTT, 100 µg/ml BSA, 0.9% Triton X-100) per 7 million cells starting material. After ligation, reverse crosslinking (65°C overnight in the presence of Proteinase K (Roche)) was followed by RNase A (Roche) treatment and two sequential phenol/chloroform extractions. After DNA precipitation (sodium acetate 3 M pH 5.2 (1/10 volume) and ethanol (2.5 x volumes)) overnight at −20°C, the DNA was spun down (centrifugation 3200 x g for 30 min at 4°C). The pellets were resuspended in 400 µl TLE (10 mM Tris-HCl pH 8.0; 0.1 mM EDTA), and transferred to 1.5 ml eppendorf tubes. After another phenol/chloroform extraction and DNA precipitation overnight at −20°C, the pellets were washed three times with 70% ethanol, and the DNA concentration was determined using Quant-iT Pico Green (Life Technologies). The efficiency of biotin incorporation was assayed by amplifying a 3C ligation product (primers available upon request), followed by digest with *HindIII* or *NheI*.

To remove biotin from non-ligated fragment ends, 40 µg of Hi-C library DNA were incubated with T4 DNA polymerase (NEB) for 4 hr at 20°C, followed by phenol/chloroform purification and DNA precipitation overnight at −20°C. After a wash with 70% ethanol, sonication was carried out to generate DNA fragments with a size peak around 400 bp (Covaris E220 settings: duty factor: 10%; peak incident power: 140W; cycles per burst: 200; time: 55 s). After end repair (T4 DNA polymerase, T4 DNA polynucleotide kinase, Klenow (all NEB) in the presence of dNTPs in ligation buffer (NEB)) for 30 min at room temperature, the DNA was purified (Qiagen (Hilden, Germany) PCR purification kit). dATP was added with Klenow exo- (NEB) for 30 min at 37°C, after which the enzyme was heat-inactivated (20 min at 65°C). A double size selection using AMPure XP beads (Beckman Coulter, Brea, CA) was performed: first, the ratio of AMPure XP beads solution volume to DNA sample volume was adjusted to 0.6:1. After incubation for 15 min at room temperature, the sample was transferred to a magnetic separator (DynaMag-2 magnet; Life Technologies), and the supernatant was transferred to a new eppendorf tube, while the beads were discarded. The ratio of AMPure XP beads solution volume to DNA sample volume was then adjusted to 0.9:1 final. After incubation for 15 min at room temperature, the sample was transferred to a magnet (DynaMag-2 magnet; Life Technologies). Following two washes with 70% ethanol, the DNA was eluted in 100 µl of TLE (10 mM Tris-HCl pH 8.0; 0.1 mM EDTA). Biotinylated ligation products were isolated using MyOne Streptavidin C1 Dynabeads (Life Technologies) on a DynaMag-2 magnet (Life Technologies) in binding buffer (5 mM Tris pH8, 0.5 mM EDTA, 1 M NaCl) for 30 min at room temperature. After two washes in binding buffer and one wash in ligation buffer (NEB), PE adapters (Illumina, San Diego, CA) were ligated onto Hi-C ligation products bound to streptavidin beads for 2 hr at room

temperature (T4 DNA ligase NEB, in ligation buffer, slowly rotating). After washing twice with wash buffer (5 mM Tris, 0.5 mM EDTA, 1 M NaCl, 0.05% Tween-20) and then once with binding buffer, the DNA-bound beads were resuspended in a final volume of 90 µl NEBuffer 2. Bead-bound Hi-C DNA was amplified with seven PCR amplification cycles using PE PCR 1.0 and PE PCR 2.0 primers (Illumina). After PCR amplification, the Hi-C libraries were purified with AMPure XP beads (Beckman Coulter). The concentration of the Hi-C libraries was determined by Bioanalyzer profiles (Agilent Technologies, Santa Clara, CA) and qPCR (Kapa Biosystems (Wilmington, MA)), and the Hi-C libraries were paired-end sequenced (HiSeq 1000, Illumina) at the Babraham Institute Sequencing Facility.

For Promoter Capture Hi-C, 500 ng of Hi-C library DNA was resuspended in 3.6 µl $H_2O$, and custom hybridization blockers (Agilent Technologies) were added to the Hi-C DNA. Hybridization buffers and the custom-made RNA capture bait system (Agilent Technologies; designed as previously described (*Mifsud et al., 2015*): 37,608 biotinylated RNAs targeting the ends of 22,076 promoter-containing *HindIII* restriction fragments) were prepared according to the manufacturer's instructions (SureSelect Target Enrichment, Agilent Technologies). The Hi-C library DNA was denatured for 5 min at 95°C, and then incubated with hybridization buffer and the RNA capture bait system at 65°C. After 24 hr incubation at 65°C, biotin/streptavidin pulldown (MyOne Streptavidin T1 Dynabeads; Life Technologies) and washes were performed according to the SureSelect Target enrichment protocol (Agilent Technologies). After the final wash, the beads were resuspended in 30 µl NEBuffer 2. After a post-capture PCR (four amplification cycles using Illumina PE PCR 1.0 and PE PCR 2.0 primers), the Promoter CHi-C libraries were purified with AMPure XP beads (Beckman Coulter). The concentration of the Promoter CHi-C libraries was determined by Bioanalyzer profiles (Agilent Technologies) and qPCR (Kapa Biosystems), and the Promoter CHi-C libraries were paired-end sequenced (HiSeq 1000, Illumina) at the Babraham Institute Sequencing Facility.

## Hi-C analysis and the definition of TADs, TAD boundaries and compartments

Raw sequencing reads were processed using the HiCUP pipeline (*Wingett et al., 2015*), which mapped sequencing read pairs against the human genome (GRCh37), filtered out experimental artifacts such as circularized reads and re-ligations, and removed all duplicate read pairs. The aligned Hi-C data were analysed using HOMER v4.7 (http://homer.salk.edu/homer/) (*Heinz et al., 2010*). Coverage- and distance-related correction factors of the binned data were computed at 25 kb and 250 kb resolutions, based on the iterative correction algorithm (*Imakaev et al., 2012*). TADs were identified based on directionality indices (*Dixon et al., 2012*) of Hi-C interactions 1 Mb upstream and downstream from a 25 kb sliding window every 5 kb steps, which were then smoothed using a running average over a ± 25 kb window. TADs were called between pairs of consecutive local maxima (TAD start) and minima (TAD end) of the smoothed directionality indices with a standard score difference (TAD ΔZ score) above 2.0, and the TAD ends were extended outward to the genomic bins with no directionality bias. These TAD definitions were used to compute the fraction of significant PCHi-C interactions falling inside TADs, alongside TADs reported by *Dixon et al., 2015*. To assess the strength of the TAD boundaries crossed by promoter interactions, we defined a TAD boundary strength score (TADB ΔZ score) as the difference between the smoothed directionality index values at the local maximum (end of the preceding TAD) and the local minimum (start of the following TAD). Defined this way, TADB ΔZ scores (unlike the TAD ΔZ scores) do not depend on the stringency of the opposite boundary of the respective TAD.

A/B compartments were called by computing the principal components of the distance- and coverage-corrected interaction profile correlation matrix at 250 kb resolution (*Lieberman-Aiden et al., 2009*). Positive values of the principal component were aligned with H3K4me3 ChIP-seq signals for H9 human ESCs (*Rada-Iglesias et al., 2011*). For chromosomes 4 and X, we used the second principal component instead of the first, as the first component described the preferential contact pattern within chromosome arms rather than compartments. The principal component values ranged from −42 to 42. To quantify the compartment changes of significant interactions, each side of the interaction was classified as A or B compartment if the principal component of its 250 kb bin was above 10 or below −10, respectively. Interactions falling within the 250 bins that had the principal component scores between −10 and 10 were considered as falling outside either compartment.

## PCHi-C interaction calling

Interactions were called at the level of individual *HindIII* fragments using version 0.1.4 of the CHi-CAGO pipeline (*Cairns et al., 2016*) based on two biological replicates for each cell type that were normalised and combined as part of the pipeline. CHiCAGO incorporates a convolution background model, which emcompasses the 'Brownian' (real, but expected interactions) and 'technical' (assay and sequencing artefacts) components, and a weighted multiple testing correction procedure trained on interaction distance. CHiCAGO interaction scores correspond to –log-transformed, weighted p-values for each fragment read pair. A score threshold of 12 was used (equivalent to a threshold of 5 in Chicago v1.0.0+ due to a soft-thresholding procedure introduced in this version). This threshold was chosen empirically based on balancing the enrichment for chromatin marks at PIRs with the overall number of detected interactions. Additionally, interactions with scores between 11 and 12 were included in the analysis if they scored above 12 in the other cell type.

## RNA-sequencing libraries and analysis

Total RNA was extracted from ESCs and NECs using an RNeasy Mini Kit (Qiagen). Indexed mRNA-seq libraries were constructed from 500 ng total RNA using the Tru-Seq RNA Library Prep Kit v2 (Illumina). Library fragment size and concentration was determined using an Agilent Bioanalyzer 2100 and KAPA Library Quantification Kit (KAPA Biosystems). Samples were sequenced on an Illumina HiSeq as single-end libraries at the Babraham Institute Sequencing Facility. Reads were trimmed using trim galore (http://www.bioinformatics.babraham.ac.uk/projects/trim_galore/) with default parameters to remove the standard Illumina adapter sequence. Reads were mapped to the GRCh37 assembly using tophat (*Trapnell et al., 2009*). BAM files were imported to Seqmonk (http://www.bioinformatics.babraham.ac.uk/projects/seqmonk/). Raw read counts per transcript were calculated using the RNA-seq quantitation pipeline on the Ensembl v70 gene set using non-directional counts. Differential analysis of gene expression was performed using the default settings in DESeq2 (*Love et al., 2014*) without independent filtering of the results. Differentially expressed genes were called at padj < 0.05 and log2 fold change above 1.5 or below −1.5.

## ChIP data analysis and definition of chromatin states

The histone modification ChIP-seq data (H3K4me1, H3K4me3, H3K27ac and H3K27me3) for ESCs and NECs were from *Rada-Iglesias et al., 2011*, available in Gene Expression Omnibus under accession number GSE24447. Data were converted to GRCh37 using liftOver (*Kent et al., 2002*). CTCF ChIP-seq data were from ENCODE (*ENCODE Project Consortium, 2012*).

Chromatin segmentations were performed on the basis of multiple histone modification ChIP datasets using a Hidden Markov Model-based method implemented in ChromHMM (version 1.10) (*Ernst and Kellis, 2012*) with default settings. The segmentation was carried out jointly through providing 'concatenated' data for both cell types as input. The resulting 16 states were curated into four broad chromatin states based on analysing their enrichment for different histone marks (*Figure 1—figure supplement 3A*) as follows. States 1–6 characterised by the presence of H3K4me3 and/or H3K27ac, and the absence of H3K27me3, were labelled 'active'; states 7–9 showing a combination of H3K4 methylation and H3K27me3 were labelled 'poised'; state 10 showing H3K27me3 and no H3K4 methylation was labelled 'Polycomb-associated'; states 14–16 showing no detectable signal for the four tested histone modifications were labelled 'background'. In addition, two more curated states were defined, but not considered further: states 11–12 were characterised by a 'mixed' pattern of both H3K27ac and H3K27me3, which likely arose from a technical issue such as heterogeneity within the samples; state 13 characterised by H3K4me1 alone was classified as 'intermediate enhancers', but the fraction of PIRs bearing this signature (~1%) was too small to analyse them as an individual category. *HindIII* fragments in the human genome (including baits and PIRs) were then classified according to the chromatin states detected within them. When more than one chromatin state was present, classification was resolved in the following manner: (i) any functional state (e.g. active, poised, Polycomb-associated) was prioritised above background; (ii) active, poised and Polycomb-associated states were prioritised above intermediate; (iii) poised state was prioritised above the Polycomb-associated state; (iv) active state together with any inactive state (i.e. poised or Polycomb-associated) was labelled as mixed. Based on these heuristics, we assigned a single chromatin state (including the background state) to 81% of PIRs in both cell types.

### Tissue-specific enhancer activity

Transgenic reporter assays for enhancer activity are described within the VISTA Enhancer Browser (*Visel et al., 2007*). The enhancer sequences from VISTA were overlapped with PIRs, and their putative target genes were defined according to the PCHi-C detected promoter-PIR interactions.

### Integration with TADs and INs

*HindIII* fragments were mapped to TADs defined as described above and INs obtained from *Ji et al., 2016*. Baited fragments overlapping TAD boundaries, and those mapping outside INs, were excluded from respective analyses. For each CRU, the percentage of interactions that map within the same TAD or IN was calculated and these values were collected into 12 bins, The first and last bins contained the values of 0% and 100%, respectively, and the remaining bins contained all other values split into 10% intervals. These results were compared to 1000 random permutations of CRUs across all promoter fragments performed in a manner retaining the overall CRU structure.

### Definition of CRU clusters

Each CRU was categorised according to the fraction of PIRs in the active, poised and Polycomb-associated state. These fractions were used for hierarchical clustering based on Euclidian distances (method='Euclidian' in *dist* function in R) with the average agglomeration method (method='average' in *hclust* function in R). PIRs assigned intermediate or mixed chromatin states did not contribute to the clustering procedure, in the latter case because the states of the regulatory elements interacting with target promoters within these PIRs are not identifiable. CRUs containing only mixed or intermediate PIRs were not included in the analysis.

### Definition of retained and rewired interactions

False-negative rates associated with stringent signal thresholds drive down the observed overlap between conditions and may overestimate the proportion of cell-type-specific interactions. Therefore, we applied additional criteria to identify high-confidence subsets of rewired and retained interactions based on replicate-level CHiCAGO interaction calls. First, we required that rewired interactions have scores above 12 in both biological replicates of the same cell type, and below 12 in both replicates of the other cell type. We then binned the interactions satisfying these criteria into five groups of equal size according to their interaction scores in the merged samples. Interactions belonging to the top bin in one cell type and the bottom bin in the other cell type were considered as rewired. Interactions with scores above 12 in the two replicates of both cell types were considered as retained. Applying these criteria and filtering out interactions with PIRs in the mixed and intermediate states, we obtained high-confidence sets of 1258 retained and 1153 rewired interactions that were used in the analysis.

### Data availability

Sequencing data have been deposited in Gene Expression Omnibus (GEO) with accession number GSE86821. Processed data including interaction peak calls in the WashU Genome Browser text format and RNA-seq raw read counts were deposited in the same GEO repository. CHiCAGO objects containing all detected interactions, ChromHMM segmentation data, DESeq2-processed RNA-seq data and the defitions of TADs have been made available through the Open Science Framework (http://osf.io/sdbg4).

## Acknowledgements

We thank Kristina Tabbada at the Babraham Institute Sequencing Facility, and Sarah Elderkin, Simon Andrews, Wolf Reik and members of our groups for insightful discussions. PF, PJR–G and MS are supported by the BBSRC. CV and PF are supported by the European Research Council (Advanced Grant 111608). AJC is supported by an MRC DTG Studentship (MR/J003808/1). RG–V was supported by the ERASMUS+ Program. PJR–G was supported by the Wellcome Trust (WT093736).

## Additional information

### Funding

| Funder | Grant reference number | Author |
| --- | --- | --- |
| Wellcome | WT093736 | Peter J Rugg-Gunn |
| Biotechnology and Biological Sciences Research Council | BB/J004480/1 | Paula Freire-Pritchett<br>Stefan Schoenfelder<br>Csilla Várnai<br>Steven W Wingett<br>Jonathan Cairns<br>Mayra Furlan-Magaril<br>Peter Fraser<br>Mikhail Spivakov |
| Medical Research Council | MR/J003808/1 | Amanda J Collier |

The funders had no role in study design, data collection and interpretation, or the decision to submit the work for publication.

### Author contributions

PF-P, Conceptualization, Resources, Data curation, Formal analysis, Validation, Visualization, Methodology, Writing—original draft, Writing—review and editing; SS, Conceptualization, Resources, Investigation, Visualization, Writing—original draft; CV, Resources, Formal analysis, Methodology, Writing—review and editing; SWW, Data curation, Methodology; JC, Formal analysis, Methodology, Writing—review and editing; AJC, RG-V, Validation, Investigation, Visualization; MF-M, Investigation, Methodology; CSO, Conceptualization, Resources, Funding acquisition, Methodology, Writing—review and editing; PF, Resources, Supervision, Funding acquisition, Investigation, Writing—review and editing; PJR-G, Conceptualization, Resources, Supervision, Funding acquisition, Investigation, Visualization, Methodology, Writing—original draft, Project administration, Writing—review and editing; MS, Conceptualization, Resources, Formal analysis, Supervision, Funding acquisition, Validation, Methodology, Writing—original draft, Project administration, Writing—review and editing

### Author ORCIDs

Mikhail Spivakov, http://orcid.org/0000-0002-0383-3943

## Additional files

### Supplementary files

• Supplementary file 1. The detected interactions and the chromatin states of the corresponding promoters and PIRs. The table shows the IDs and genomic coordinates (based on GRCh37 assembly) of the baited promoter fragments (protein-coding genes only) and their respective PIRs for interactions detected in either ESCs (*ESConly*), NECs (*NEConly*), or both cell types (*Both*). The chromatin states of promoters and PIRs in each cell type are listed in the corresponding *BaitState* and *PIRState* columns. See Materials and methods for details on the thresholding approach used. CHiCAGO objects containing read-count and score information for all sequenced fragment pairs in ESCs and NECs are available through Open Science Framework (http://osf.io/sdbg4).

• Supplementary file 2. PCHi-C candidate genes for enhancer regions annotated in the VISTA Enhancer Browser. The genomic coordinates (based on GRCh37 assembly) of enhancers annotated in the VISTA Enhancer Browser that map to PIRs detected in either ESCs, NECs, or both (*ESConly*, *NEConly* and *Both*), their reported tissue-specificities, and the associated PCHi-C-identified putative target genes. The columns *ActiveESC* and *ActiveNEC* list whether the corresponding VISTA enhancers overlap with active chromatin marks in ESCs and NECs, respectively.

• Supplementary file 3. The properties of the identified CRUs in ESCs and NECs. The table lists the following CRU information: associated gene name, gene expression (processed with DESeq2), number of PIRs, the promoter (bait) chromatin state, single/dual-state annotation, CRU cluster ID and

CRU chromatin state transitions between ESCs and NECs. Only CRUs which have been assigned to clusters in both ESC and NEC are listed.

## Major datasets

The following datasets were generated:

| Author(s) | Year | Dataset title | Dataset URL | Database, license, and accessibility information |
|---|---|---|---|---|
| Paula Freire-Pritchett, Stefan Schoenfelder, Csilla Várnai, Steven W Wingett, Jonathan Cairns, Amanda J Collier, Raquel García-Vílchez, Mayra Furlan-Magaril, Cameron S Osborne, Peter Fraser, Peter J Rugg-Gunn, Mikhail Spivakov | 2016 | Global rewiring of cis-regulatory units upon lineage commitment of human embryonic stem cells | https://www.ncbi.nlm.nih.gov/geo/query/acc.cgi?acc=GSE86821 | Publicly available at the NCBI Gene Expression Omnibus (accession no: GSE86821) |
| Paula Freire-Pritchett, Stefan Schoenfelder, Csilla Várnai, Steven W Wingett, Jonathan Cairns, Amanda J Collier, Raquel García-Vílchez, Mayra Furlan-Magaril, Cameron S Osborne, Peter Fraser, Peter J Rugg-Gunn, Mikhail Spivakov | 2016 | Global rewiring of cis-regulatory units upon lineage commitment of human embryonic stem cells | https://osf.io/sdbg4 | Publicly available via the Open Science Framework |

The following previously published dataset was used:

| Author(s) | Year | Dataset title | Dataset URL | Database, license, and accessibility information |
|---|---|---|---|---|
| Rada-Iglesias A, Wysocka J | 2010 | A unique chromatin signature uncovers early developmental enhancers in humans | https://www.ncbi.nlm.nih.gov/geo/query/acc.cgi?acc=GSE24447 | Publicly available at the NCBI Gene Expression Omnibus (accession no: GSE24447) |

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
