## [Decision Letter]

Thank you for submitting your article "Global reorganisation of cis-regulatory units upon lineage commitment of human embryonic stem cells" for consideration by *eLife*. Your article has been reviewed by three peer reviewers, one of whom, Job Dekker (Reviewer#1), is a member of our Board of Reviewing Editors, and the evaluation has been overseen by Fiona Watt as the Senior Editor. The following individuals involved in review of your submission have agreed to reveal their identity: Frank Alber (Reviewer #3).

The reviewers have discussed the reviews with one another and the Reviewing Editor has drafted this decision to help you prepare a revised submission. In the course of discussion the reviewers and editors decided that this work would be more appropriately reconsidered as a Tools and Resources (TR) paper rather than a Research Article (RA). The work appears to be a valuable resource but the primary research results are not considered sufficiently novel to warrant publication as an RA. Nonetheless, we hope you are willing to go forward with this in the category of a TR paper.

Summary:

Freire-Pritchett et al. employ Capture Hi-C to detect chromatin interactions between promoters and distal elements in ES cells and in ESC derived neuroectodermal cells (NECs). The authors report significant interactions between many promoters and different distal elements in ESCs and NECs. Analysis of the chromatin state of these elements indicates that these are functional elements including (poised) enhancers. Further, the connectivity between promoters and distal elements differs between ESCs and NECs and this relates to the transcriptional status of the genes and chromatin status of the distal elements. The authors then define cis-regulatory units (CRUs) as promoters and their associated distal elements. Although many of the CRUs are contained within TADs and INs, as had been expected based on earlier studies, the authors report that many can extend beyond TAD and IN boundaries.

The main impact of this paper is to announce the availability of a large data set of enhancer-promoter contacts identified via capture Hi-C. Many of the experimental technologies and statistical analysis methods are established and rely on previously published methodologies. That said the reviewers raise concerns about specific aspects of the analysis. Further, many of the reported correlations are already known and the results confirm earlier studies (e.g. promoters interact with multiple distal elements, these elements coincide with cell type-specific enhancer-like elements and these connections are equally cell type-specific, and enriched for intra-TAD interactions). The main new aspect of this work is the fact that a genome-wide promoter-anchored interaction atlas is described, and this resource could be of interest to the community given that many study these cells.

Essential revisions:

Please address these main criticisms (more details are in the minor points section):

1) Many of the correlations between looping, gene expression and chromatin state are not novel. Focus on the new things, and present the data as a large dataset or resource for the community.

2) The reviewers raised issues related to the statistical analysis and computational methods to determine PIRs (and their hierarchical clustering), TADs, and CRUs (below). Please address all of them.

3) Related to the previous point, the reviewers raised concerns about the claim that CRUs are distinct from INs, TADs. The main point to address is how the methods to determine these features compare and whether lack of overlap can be due to sensitivity of the computational methods.

Minor points:

1) One concern is that the CHiCAGO method identifies significant interactions using only the general distance decay of interactions into account. It does not take into account that any pair of interactions between loci located within a TAD or IN, or between loci located in a similar compartment (A, B or subcompartment types) are generally higher than loci located in different TADs/INs/Compartments. As a result, a number of significant interactions are not necessarily "loops" (specific point-to-point interactions). This is not to say that these loci do not interact more frequently than expected given their genomic distance, but these interactions can reflect general higher order structures such as domains etc. This can explain why many interacting loci appear to not have chromatin marks: these can be "bystander" interactions that are the result of a nearby structural feature. I recommend that the authors repeat their analysis using a background model that takes domains into consideration.

2) The TAD calling procedure is quite simple. Is there any evidence that this procedure produces TAD calls of comparable quality to those produced by, for example, the HMM of Dixon et al. Nature 2012?

3) The TAD and IN comparison is unsatisfying. The main conclusion seems to be that CRUs are not particularly aligned with TADs and INs. This could be because the two phenomena are more or less independent of each other, or because the TADs, INs and CRUs are each defined in a fairly arbitrary fashion. If, for example, the simple TAD calling scheme employed here is not working well, then it would lead to the observed discordance.

4) The finding that CRUs can extend beyond TAD or IN boundaries is interesting, but the data is not sufficient to claim a new "feature of genome architecture". Further analyses are needed such as: when a CRU extends beyond a TAD boundary, is this a weak TAD boundary? It is well known that TADs are nested structures. Do CRUs remain within the larger nested domains? This seems to be the case for POUF2. Quantification of boundary strength is required, as it seems not inconsistent with the TAD/IN models of enhancer action if the authors find interactions across weak boundaries, but not strong boundaries. Also, are these interactions beyond the boundaries equally strong, or are they much lower in contact frequency

5) The main text claims that the promoter capture Hi-C data was processed to identify "significant cis-interactions." Looking at Methods, we learn that these contacts correspond to a threshold of 12, on a "log-transformed, weighted p-value" scale. This vague description makes it impossible to actually interpret the confidence associated with a threshold of 12, especially since we are told that the threshold corresponds to "a threshold of 5 in Chicago v1.0.0+ due to a soft-thresholding procedure introduced in this version." The choice of threshold must be fully justified and described. The text claims that Chicago uses an FDR control procedure, so the interactions that are presented here should be reported along with their FDR threshold.

6) The manuscript contains no details on precisely how ChromHMM was run. Details like software version numbers and any parameter settings should be given. Importantly, the text should specify whether the segmentation was carried out independently on each cell type, or jointly, and if the latter, whether the data tracks were "stacked" (i.e., 8 marks per position) or "concatenated" (4 marks per position, across two concatenated copies of the genome).

7) Details of the "curation" of the 16 inferred states should also be provided. When the 16 states were collapsed to four states, what was the basis for this collapsing? It seems strange that some states were eliminated at this stage because they contained multiple marks.

8) How are PIRs defined: at the level of individual restriction fragments, or are adjacent fragments that both score significant merged into a single PIR?

9) The authors do not show many tracks of real data (only in one of the supplemental figures). Throughout they only show arcs to indicate significant contacts. This makes it difficult to assess the quality of the data.

10) Promoters often interact with CTCF sites. Did the authors find this in their dataset as well?

11) The authors focus on cis-interactions. Can the authors describe trans interactions as well? Given that trans-regulation is considered rare, this analysis will provide context for interpretation of the cis data.

12) Did the authors detect any known enhancer-promoter pairs? Also, the authors imply that the distal elements that touch promoters are regulating these promoters, but beside correlations between interactions, transcription and chromatin state of the distal elements, no evidence for direct regulation is provided. This would require deletion of the elements, e.g. by CRISPR. The Vista analysis of the POUF2 interactions suggests that many PIRs are enhancers. That is similar in information as the observation that PIRs overlap loci with histone modifications associated with enhancers. The key thing to validate is that these enhancers indeed regulate the interacting promoter. The fact that the pattern of activity of these enhancers resembles POU3F2 expression is encouraging but not definitive. The authors should either provide direct evidence for functional relationships or clearly discuss the extent to which the current data predicts such relationships even though the data currently is correlative.

13) The authors examine the dynamic rewiring and recoloring of interactions as cells differentiate from ESCs to NECs. They report gain and loss of interactions and these are related to gain or loss of chromatin marks. This is interesting, but what needs to be tested is whether this is related compartment changes (see above). This is not to say these interactions are not relevant, but it would be important to determine how they relate to larger scale compartment changes. One interesting implication could be that in fact cell type-specific compartment changes are driven by altered interactions between functional elements that are active in a cell type-specific manner.

14) How do the authors interpret interactions that involve non-expressed genes?

15) The hierarchical clustering of PIRs by the prevalence of different chromatin state labels is fairly uninformative. The fact that these different categories exist in the data is not surprising, and the claim that these eight labels somehow have "potential implications for understanding the logic of signal integration at promoters" is not well supported.

16) In the Abstract, the sentence "Here, we generate[…]" should mention what assay was employed.

17).In the Introduction: "The extreme combinatorial complexity of Hi-C[…]" The complexity is really only quadratic, not combinatorial.

18). In the Results section: The URL for data availability should be included at the end of the sentence, "This data resource[…]"

19) Also in the Results section: "NEC PIRs were strongly enriched" This claim requires statistical support.

20) New paragraph at "We next sought[…]"

21) Two of the observations offered in the Discussion do not seem to be supported by direct evidence in Results: "we find extensive promoter connectivity to regions associated with Polycomb-associated repression and poising, and "we[…] detect large numbers of promoter interactions with regions devoid of chromatin features." These observations seem to be offered for the first time in Discussion.

22) The work does not show whether all the interactions are present simultaneously in a cell, or if the promoter state may vary between cells, depending on the probability with which the promoter is interacting with any of the potential alternative enhancers of a given chromatin state. Please discuss this issue in the main text.

23) Out of the ~21000 promoters, the authors focused their CRU analysis on only 16,000 protein-coding genes. Would the authors expect differences in the outcome if the other remaining promoters would be considered?

24) Out of the 16,000 protein-coding promoters about 9000 were defined as CRU. It may be good to know the selection criteria (i.e. is it based on a minimum number of PIRs per promoter?).

25) Figure 3 defines a CRU as a set of PIRs that seem to be concurrently interacting with the promoter in the same cell. Maybe the authors could indicate in the caption that it cannot be ruled out that some PIRs may provide alternative rather then concurrent interactions.

---

## [Author Response]

*Essential revisions:*

*Please address these main criticisms (more details are in the minor points section):*

*1) Many of the correlations between looping, gene expression and chromatin state are not novel. Focus on the new things, and present the data as a large dataset or resource for the community.*

We thank the editors and reviewers for their constructive and thorough feedback on our study. We are happy for our work to be reconsidered as a Tools and Resources paper and we have taken on board the advice to focus more on the resource aspect of our study. We have achieved this, in part, by describing the large dataset more fully, adding further methodological details and support, and adopting a more cautious posture regarding the correlations between interactions and chromatin state.

*2) The reviewers raised issues related to the statistical analysis and computational methods to determine PIRs (and their hierarchical clustering), TADs, and CRUs (below). Please address all of them.*

*3) Related to the previous point, the reviewers raised concerns about the claim that CRUs are distinct from INs, TADs. The main point to address is how the methods to determine these features compare and whether lack of overlap can be due to sensitivity of the computational methods.*

We thank the editors and reviewers for their helpful feedback on these two points. We address these criticisms in our responses below.

*Minor points:*

*1) One concern is that the ChICAGO method identifies significant interactions using only the general distance decay of interactions into account. It does not take into account that any pair of interactions between loci located within a TAD or IN, or between loci located in a similar compartment (A, B or subcompartment types) are generally higher than loci located in different TADs/INs/Compartments. As a result, a number of significant interactions are not necessarily "loops" (specific point-to-point interactions). This is not to say that these loci do not interact more frequently than expected given their genomic distance, but these interactions can reflect general higher order structures such as domains etc. This can explain why many interacting loci appear to not have chromatin marks: these can be "bystander" interactions that are the result of a nearby structural feature. I recommend that the authors repeat their analysis using a background model that takes domains into consideration.*

The reviewers are correct to point out that CHiCAGO’s domain-unaware model somewhat overestimates the background read count (i.e. that expected at random) for TAD-crossing interactions (Figure 7). As a result of this, CHiCAGO is relatively more stringent (conservative) in calling TAD-crossing interactions than within-TAD interactions. In practice, the detected interactions are likely to be of comparable strength, as evidenced by the very similar read counts at the detected within-TAD and TAD-crossing interactions at a given interaction distance range (Figure 7). Therefore, the key observation of our study that TADs constrain, but do not fully insulate, promoter interactions is unaffected: the TAD-unaware CHiCAGO background only pushes us to err on the side of caution in detecting TAD-crossing interactions.

In the future, it would indeed be worthwhile to incorporate domain information into the CHiCAGO background model. However, this is a complex and challenging task. One of the challenges that need to be addressed, for example, is that the background depends not just on whether an interaction crosses a TAD boundary, but also on TAD boundary stringency, and does so in a way that scales non-linearly with distance (Figure 7). It is also likely that the background is affected by the number of crossed boundaries. Modelling all of these dependencies in the data must be performed with great care to avoid the danger of overfitting, which could lead to circular reasoning when attempting to make inferences about TADs.

Since the identified TAD-crossing interactions are at least no weaker than the within-TAD interactions, and since implementing TAD-aware interaction calling is likely a challenging project of its own, we feel that addressing this point in full goes outside the scope of this study.

Author response image 1.CHiCAGO detection of cross-TAD and within-TAD interactions.(**A**) The effect of TAD boundary crossing on background levels. In line with the CHiCAGO background estimation procedure (see Additional file 1, Cairns et al., 2016), we used 20kb distance bins to tile 3Mb regions centred at each bait. For each bin and bait, we computed a normalized number of reads as the average number of reads linking fragments in this bin with the baited fragment, divided by the baited fragment’s bias factor. These values were then considered separately for bins mapping within the same TAD as the respective bait and for those separated from the bait by at least one TAD boundary. Solid lines show the median log-transformed normalised number of reads and ribbons show the interquartile range. It can be seen that the CHiCAGO background estimate (dashed line) is very close to the background estimate computed for within-TAD interactions separately, suggesting that within-TAD pairs primarily drive CHiCAGO’s background model. In contrast, bins that lie beyond a TAD boundary have lower background levels, consistent with the reviewers’ expectation. Furthermore, stronger TAD boundaries correspond to a larger decrease in background level, though this effect is nonlinear. Since the estimated background is higher than the true background, this increases the stringency of CHiCAGO calls for TAD-crossing interactions. (**B**) The read counts of significant interactions, stratified by whether they cross a TAD boundary or not, and by distance bin. It can be seen the read count distributions of the CHiCAGO-identified TAD-crossing and within-TAD interactions (CHiCAGO score > 12) show only minor differences.**DOI:**
http://dx.doi.org/10.7554/eLife.21926.022

Author response image 2.Tuning a signal threshold for TAD detection with HOMER.(**A**) Box plots show TAD length distributions for TADs called by HOMER using a range of TAD ∆Z score thresholds (blue to red boxes), as well as for the published TADs from Dixon et al., 2015 detected with individual and combined Hi-C replicates (green boxes). The dashed grey line marks the median TAD length of the combined Dixon et al., 2015 data. (**B**) Overlap with CTCF sites for TADs called by HOMER using a range of TAD ∆Z score thresholds (blue to red lines), and for the published TADs from Dixon et al., 2015. (green lines). Solid lines show the observed fraction of TAD boundaries mapping within given distances from CTCF sites, and dashed lines show the expected fraction using boundaries shifted by +1 Mb for each set of boundaries. (**C**) TAD boundary overlap between the TADs called with HOMER at ∆Z > 2.0 and the Dixon et al., 2015 et al. TADs (combined replicates). The bars show the numbers of HOMER TADs with zero, one or two TAD boundaries within 50 kb of a Dixon et al., 2015 TAD boundary, respectively. (**D**) A WashU browser view of the directionality index profile and the locations of TADs called in an example ~20 Mb region on chromosome 7. Top row: Directionality indices (shades of blue and red represent the negative and positive index ranges, respectively). Middle row (green): Dixon et al., 2015 TADs (combined replicates). Bottom row (blue): HOMER TADs called at TAD ∆Z > 2.0.**DOI:**
http://dx.doi.org/10.7554/eLife.21926.023

*2) The TAD calling procedure is quite simple. Is there any evidence that this procedure produces TAD calls of comparable quality to those produced by, for example, the HMM of Dixon et al. Nature 2012?*

We agree that a comparison with Dixon et al., 2015 data is worthwhile and have now performed this analysis. This has revealed that the cutoff on HOMER boundary stringency scores that we initially used (∆Z>1) likely results in a more lenient TAD definition than the Dixon et al., 2015 HMM-based method, as evidenced by the higher numbers of detected TADs (4491 vs 3062) and smaller TAD sizes (Figure 8). Notably, the boundaries used in the initial submission showed a comparable enrichment for CTCF binding sites to Dixon et al., 2015 boundaries (Figure 8).

In the absence of a gold standard for TAD detection, it is difficult to judge which of the two methods is “more correct”. However, for consistency we now use a more stringent HOMER threshold (∆Z>2) that results in fewer TADs (2761 vs 4491 with ∆Z>1) and a comparable distribution of TAD lengths with the Dixon et al., 2015 TADs (Figure 8). More than 70% of our called TADs have at least one boundary that is consistently detected in the Dixon et al. study (Figure 8). To illustrate the boundaries called with both methods, we show their TAD calls for an example region in Figure 8.

The figures and the main text of the manuscript have been updated based on the new set of TAD calls. Using the more stringent set of TADs has resulted in fewer CRUs identified as crossing TAD boundaries (just over 20% vs ~40% in the initial set). However, this is primarily due to the larger sizes of the updated TADs rather than their higher ‘insulation power’. The ratio of observed-to-expected numbers of CRUs fully contained within TAD boundaries has in fact gone slightly down in the updated set (from 1.5-fold to 1.3-fold). We have also added the results for Dixon et al., 2015 TADs to Figure 3—figure supplement 1, to enable a direct comparison. These data show that a similar proportion of CRUs cross Dixon et al., 2015 TAD boundaries (~27%) compared to those in our updated HOMER-defined set (~20%; ∆Z>2), demonstrating that the overall conclusion of this analysis is robust across at least two commonly used TAD-calling methods.

*3) The TAD and IN comparison is unsatisfying. The main conclusion seems to be that CRUs are not particularly aligned with TADs and INs. This could be because the two phenomena are more or less independent of each other, or because the TADs, INs and CRUs are each defined in a fairly arbitrary fashion. If, for example, the simple TAD calling scheme employed here is not working well, then it would lead to the observed discordance.*

We have now modified the TAD analysis as described above, which we believe addresses this point as far as TADs are concerned.

The IN calls have been taken verbatim from the study that introduced this concept (Ji et al., 2016). We noted, however, that the reported INs were often nested, so we have repeated the analysis using only the largest span of each overlapping set. As expected from the larger IN sizes, the numbers of IN-crossing CRUs went down somewhat, but still less than half of them (~45%) were fully contained within the extremities of IN boundaries (Figure 3—figure supplement 1). Therefore, these results further strengthen a principal conclusion of our analysis that while INs constrain promoter interactions to some extent, they do not fully insulate them.

*4) The finding that CRUs can extend beyond TAD or IN boundaries is interesting, but the data is not sufficient to claim a new "feature of genome architecture". Further analyses are needed such as: when a CRU extends beyond a TAD boundary, is this a weak TAD boundary? It is well known that TADs are nested structures. Do CRUs remain within the larger nested domains? This seems to be the case for POUF2. Quantification of boundary strength is required, as it seems not inconsistent with the TAD/IN models of enhancer action if the authors find interactions across weak boundaries, but not strong boundaries. Also, are these interactions beyond the boundaries equally strong, or are they much lower in contact frequency*

To address this question, we plotted the distributions of HOMER strength scores for TAD boundaries that are crossed and not crossed by promoter interactions (Figure 3—figure supplement 1). As can be seen, TAD boundaries crossed by interactions are indeed slightly weaker overall (Wilcoxon test p-value = 1.8e-14). However, the ranges fully overlap, and even the strongest TAD boundaries are not fully impenetrable to promoter interactions. We now also mention this analysis in the main text. Although these new findings have strengthened our analysis of CRUs, we take on board the reviewers’ concern about our interpretation of CRUs as new features of genome architecture, and have removed it from the manuscript.

As for the contact frequency of TAD-crossing and within-TAD interactions, it can be seen from Figure 7 that it is very similar in both cases given the distance. In the short range, the identified TAD-crossing interactions even appear to have a slightly higher read count, most likely owing to the domain-unaware CHiCAGO background model, as discussed in our response to minor point 1.

Finally, we agree that empirically, TADs indeed represent nested structures, and tools for TAD calling in a hierarchical fashion are beginning to emerge (although neither the Dixon et al. method nor HOMER are among them). However, the conceptual disparity between the different levels of TAD “nesting” is still unclear. For example, from what level of the hierarchy should a sub-TAD be a considered a TAD, or a TAD be considered a “super-TAD”, etc.? It is possible to devise criteria that seem intuitively acceptable, but it will take time for the underlying assumptions of such criteria to acquire a solid structural and biophysical underpinning.

*5) The main text claims that the promoter capture Hi-C data was processed to identify "significant cis-interactions." Looking at Methods, we learn that these contacts correspond to a threshold of 12, on a "log-transformed, weighted p-value" scale. This vague description makes it impossible to actually interpret the confidence associated with a threshold of 12, especially since we are told that the threshold corresponds to "a threshold of 5 in Chicago v1.0.0+ due to a soft-thresholding procedure introduced in this version." The choice of threshold must be fully justified and described. The text claims that Chicago uses an FDR control procedure, so the interactions that are presented here should be reported along with their FDR threshold.*

We believe that, in part, this question has arisen because we made a confusing statement in the text. CHiCAGO’s weighted multiple testing treatment is not, strictly speaking, “a false discovery control procedure”, and we have now altered the main text to clarify this.

More generally, multiple testing correction when calling peaks in genomic data is a challenging problem, leading to the development of alternatives to standard approaches such as Bonferroni and FDR (such as, for example, IDR for ChIP-seq data). In PCHi-C data, an additional challenge is presented by undersampling, which leads to the violation of the underlying assumptions of both the IDR and conventional FDR approaches. In the case of FDR, this is because the majority of fragment pairs have only one or zero reads, leading to highly non-uniform p-value distributions. Furthermore, in PCHi-C data false discovery rates strongly vary with linear distance, partly due to the decreasing statistical power as distance increases. To account for this property, CHiCAGO adjusts the p-values using a weighting procedure, in which the weights trained on the reproducibility of interaction calls with distance (see Cairns et al., 2016 for details). Soft-thresholding was further introduced in CHiCAGO v1.0.0+ to shift the weighted -log-p-values (scores) such that the score of zero corresponds to the probability of a true interaction in the very short range given zero reads.

CHiCAGO scores are conceptually similar, albeit clearly not identical, to posterior probabilities resulting from an empirical Bayesian treatment. We view them primarily as a ranking measure, and use a heuristic approach based on integration with other types of genomic data to choose a threshold. Specifically, for a range of threshold settings, we compute the observed overlap of the identified PIRs with genomic regulatory features (such as specific histone marks) and the same overlap expected at random (estimated with random regions that are promoter distance-matched to the detected PIRs). In choosing a threshold, we balance the ‘unexpected’ overlap of PIRs with regulatory features with the total number of interactions detected at a given setting. As can be seen in Figure 9, at the selected threshold of 12 the relative difference in the overlap of PIRs with histone marks and CTCF versus random controls begins to saturate. Further increasing the threshold leads to a continued drop in the identified interactions (Figure 9) without a considerable further gain in the ‘unexpected’ overlap of PIRs with regulatory features.

We now explicitly mention the rationale for choosing the threshold in the Methods section. However, we agree that it is a crude and potentially subjective method. While varying the threshold within the reasonable range is unlikely to affect the principal conclusions of our study, we release the scores for each fragment pair as part of the Chicago objects on Open Science Framework, so the users can employ other methods of tuning the threshold, as well as apply non-threshold-based approaches to analysing our data.

Author response image 3.Properties of CHiCAGO interactions called at different score thresholds in ESCs and NECs.(**A** and **B**) The relative difference between observed and expected numbers of PIRs overlapping with the ChIP-seq peaks of histone modifications and CTCF, detected at a range of CHiCAGO score thresholds in ESCs (**A**) and NECs (**B**). The expected numbers of PIRs overlapping with ChIP peaks were computed in the same way as in Figure 1, as the average overlap obtained with 100 draws of promoter distance-matched random regions. (C and D) The fraction of fragment pairs with at least one PCHi-C read mapping to them that have CHiCAGO scores passing a given threshold in ESCs (**C**) and NECs (**D**).**DOI:**
http://dx.doi.org/10.7554/eLife.21926.024

*6) The manuscript contains no details on precisely how ChromHMM was run. Details like software version numbers and any parameter settings should be given. Importantly, the text should specify whether the segmentation was carried out independently on each cell type, or jointly, and if the latter, whether the data tracks were "stacked" (i.e., 8 marks per position) or "concatenated" (4 marks per position, across two concatenated copies of the genome).*

We apologise for this omission and have now added full details on the chromHMM procedure to the Methods section. Briefly, the segmentation was carried out jointly, and the data tracks were concatenated.

*7) Details of the "curation" of the 16 inferred states should also be provided. When the 16 states were collapsed to four states, what was the basis for this collapsing? It seems strange that some states were eliminated at this stage because they contained multiple marks.*

We have added full details on how the chromHMM states were curated to the Methods section. Briefly, the states were curated into “Active”, “Poised”, “Polycomb-associated” and “Background” based on the presence of active (H3K4me1/H3K27ac) and repressive histone (H3K27me3) marks. Overall, we were able to unambiguously assign one of these chromatin states to 81% of PIRs in both cell types. Two more curated states were defined, but not considered further:

1) “intermediate enhancers” characterised by H3K4me1 alone; the fraction of PIRs bearing this signature (~1%) was too small to analyse them as an individual category, and yet we felt that merging them with any other category was not biologically appropriate;

2) the ambiguous (“mixed”) state that showed enrichment for both H3K27ac and H3K27me3; we felt that this conflicting pattern likely arose from a technical issue such as heterogeneity within the samples.

*8) How are PIRs defined: at the level of individual restriction fragments, or are adjacent fragments that both score significant merged into a single PIR?*

PIRs are defined at the level of individual restriction fragments, with no signal smoothing or pooling across multiple sites. We have now clarified this in the Methods section. We have opted against further peak processing, because at present there is no reliable way to “fine-map” interaction signals without making *a priori* assumptions about the “causal” regions driving the signals, which we feel still require further mechanistic support at this stage.

*9) The authors do not show many tracks of real data (only in one of the supplemental figures). Throughout they only show arcs to indicate significant contacts. This makes it difficult to assess the quality of the data.*

We now provide two additional figures (Figure 1—figure supplement 4 and Figure 4—figure supplement 3), in which we plot the raw data for all examples shown in this study.

*10) Promoters often interact with CTCF sites. Did the authors find this in their dataset as well?*

We have now incorporated this analysis. Indeed, PIRs are highly enriched for CTCF binding sites, as we now show using ENCODE CTCF ChIP-seq data for hESCs in Figure 1—figure supplement 2, and describe in subsection “Identification of putative regulatory elements and their associated gene promoters” of the main text.

*11) The authors focus on cis-interactions. Can the authors describe trans interactions as well? Given that trans-regulation is considered rare, this analysis will provide context for interpretation of the cis data.*

We now describe in the main text that we detect 338 trans-interactions in ESCs and 266 in NECs (representing <0.5% of total interactions). Consistent with our previous observations in other PCHi-C datasets (Cairns et al., 2016)(, a large proportion of trans-interactions (50.3% and 64.3%, respectively) reflect large-scale compartment signal that is detectable in conventional Hi-C data at 1-Mb resolution for the same cell types (data not shown). Given the small numbers of detected trans-interactions, we felt we should focus on cis-interactions in the manuscript. However, we agree that trans-interactions may be useful for future reanalyses of our data, and report them in the data files released on Open Science Framework.

*12) Did the authors detect any known enhancer-promoter pairs? Also, the authors imply that the distal elements that touch promoters are regulating these promoters, but beside correlations between interactions, transcription and chromatin state of the distal elements, no evidence for direct regulation is provided. This would require deletion of the elements, e.g. by CRISPR. The Vista analysis of the POUF2 interactions suggests that many PIRs are enhancers. That is similar in information as the observation that PIRs overlap loci with histone modifications associated with enhancers. The key thing to validate is that these enhancers indeed regulate the interacting promoter. The fact that the pattern of activity of these enhancers resembles POU3F2 expression is encouraging but not definitive. The authors should either provide direct evidence for functional relationships or clearly discuss the extent to which the current data predicts such relationships even though the data currently is correlative.*

This is an important point. We have added new text in the Discussion to highlight that our data set is predictive of regulatory relationships, but that extensive functional validation is required to infer direct regulation. In the absence of a comprehensive “VISTA-style” catalog of functionally validated enhancer-promoter relationships, we hope that large-scale resources such as ours will provide a strong impetus for future validation studies.

*13) The authors examine the dynamic rewiring and recoloring of interactions as cells differentiate from ESCs to NECs. They report gain and loss of interactions and these are related to gain or loss of chromatin marks. This is interesting, but what needs to be tested is whether this is related compartment changes (see above). This is not to say these interactions are not relevant, but it would be important to determine how they relate to larger scale compartment changes. One interesting implication could be that in fact cell type-specific compartment changes are driven by altered interactions between functional elements that are active in a cell type-specific manner.*

We have now done this analysis. While ~2% of 1-Mb regions transition between A and B compartments upon differentiation (in either direction), nearly all high-confidence rewired interactions detected in our data (>99%) map outside of these “dynamic” regions. We now mention this observation in the main text.

14) How do the authors interpret interactions that involve non-expressed genes?

This is an interesting point, which we now present more extensively in the Discussion. There are several possibilities to interpret interactions with non-expressed gene promoters. For example, it is known that at least in mouse ESCs, inactive genes and their regulatory regions form extensive topological networks (see, e.g., Schoenfelder et al., 2015b ). However, we do not intend to imply that all promoter interactions have a gene regulatory nature: some of them are potentially structural. Furthermore, additional and not yet fully understood mechanisms of topologically-mediated gene repression and priming may also be at play, particularly given the fact that a number of regions unmarked by conventional chromatin signatures currently emerge as having roles in gene regulation (Pradeepa et al., 2016; Rajagopal et al., 2016).

*15) The hierarchical clustering of PIRs by the prevalence of different chromatin state labels is fairly uninformative. The fact that these different categories exist in the data is not surprising, and the claim that these eight labels somehow have "potential implications for understanding the logic of signal integration at promoters" is not well supported.*

We agree that the majority of CRUs having consistent chromatin marks between PIRs and the promoter is relatively unsurprising. However, we feel that the classification we provide will be helpful for the users of our resource to make the most out of our data, and in particular to select targets for candidate-based studies. Additionally, our hierarchical clustering approach has been instrumental for detecting the relatively large numbers of ‘inconsistent’ CRUs, leading to the observation that the chromatin states at their promoters generally follow the ‘majority rule’ with respect to the chromatin states of the PIRs. We believe that these findings lead to specific and testable mechanistic hypotheses, and are therefore also interesting in this respect. Nevertheless, we concede that our claim about “potential implications[…]” was perhaps too far-fetched, so we have removed this sentence from the manuscript.

*16) In the Abstract, the sentence "Here, we generate[…]" should mention what assay was employed.*

This has now been done.

*17) In the Introduction: "The extreme combinatorial complexity of Hi-C[…]" The complexity is really only quadratic, not combinatorial.*

We have removed the words “extreme combinatorial” from this sentence to avoid confusion.

*18) In the Results section: The URL for data availability should be included at the end of the sentence, "This data resource[…]"*

We have now inserted a new sentence with details on the availability of raw and processed data.

*19) Also in the Results section: "NEC PIRs were strongly enriched" This claim requires statistical support.*

In the initial submission, the enrichment z-scores were given in Figure 2—figure supplement 1, but not mentioned in the main text. We have now modified the text to rectify this.

*20) New paragraph at "We next sought.[…]"*

We have amended this.

*21) Two of the observations offered in the Discussion do not seem to be supported by direct evidence in Results: "we find extensive promoter connectivity to regions associated with Polycomb-associated repression and poising, and "we.… detect large numbers of promoter interactions with regions devoid of chromatin features." These observations seem to be offered for the first time in Discussion.*

Thank you for drawing our attention to this. We have now added details on these two points into the second section of Results (“Identification of putative regulatory elements and their associated gene promoters”).

*22) The work does not show whether all the interactions are present simultaneously in a cell, or if the promoter state may vary between cells, depending on the probability with which the promoter is interacting with any of the potential alternative enhancers of a given chromatin state. Please discuss this issue in the main text.*

This is a very good point. Indeed, Hi-C-based methods cannot distinguish between these two possibilities, and there is currently evidence in support of both the “hit-and-run” and the “chromatin hub” models. We discuss these topics in the section “Implications for developmental gene regulation by multiple enhancers”, which we have now expanded to explicitly outline these points.

*23) Out of the ~21000 promoters, the authors focused their CRU analysis on only 16,000 protein-coding genes. Would the authors expect differences in the outcome if the other remaining promoters would be considered?*

We now show the results for all captured promoters in Figure 1, and have updated the main text accordingly. As expected, PIR enrichment for chromatin marks and the consistency between the promoter and PIR chromatin states are nearly identical when computed for protein-coding versus all captured promoters. Because of the level of similarity, we would also not expect any meaningful differences at the CRU level for all promoters compared to the currently analysed set of protein-coding genes. We would prefer to focus our analysis on protein-coding CRUs, however, as our mRNA-seq data interrogates only coding transcripts.

*24) Out of the 16,000 protein-coding promoters about 9000 were defined as CRU. It may be good to know the selection criteria (i.e. is it based on a minimum number of PIRs per promoter?).*

We apologise for not explaining this clearly. The ~9,000 figure includes all CRUs that have at least one PIR per promoter. We have updated the main text to explicitly state this fact.

*25) Figure 3 defines a CRU as a set of PIRs that seem to be concurrently interacting with the promoter in the same cell. Maybe the authors could indicate in the caption that it cannot be ruled out that some PIRs may provide alternative rather then concurrent interactions.*

We agree that this is a good idea and have amended the legend of Figure 3 accordingly.